# VIDEOCOGQA: A CONTROLLABLE BENCHMARK FOR EVALUATING COGNITIVE ABILITIES IN VIDEO-LANGUAGE MODELS

## ABSTRACT

Recent advances in Large Video-Language Models (LVLMs) have led to promising results in multimodal video understanding. However, it remains uncertain whether these models possess the key cognitive capabilities for high-level tasks, especially those requiring symbolic and abstract reasoning. Existing benchmarks predominantly rely on real-world, annotated videos, which suffer from a lack of control over content and inherent difficulty, limiting their diagnostic utility. To address these limitations, we introduce **VideoCogQA**, a scalable and fully controllable benchmark inspired by game-based environments, designed to assess the cognitive abilities of LVLMs. By generating synthetic videos through a programmatic engine, VideoCogQA offers precise control over visual elements, temporal dynamics, and the task difficulty, effectively isolating cognitive reasoning from prior semantic knowledge. The dataset consists of tasks involving abstract concepts, symbolic elements, and multimodal integration, with varying levels of difficulty based on Python-based game scenarios. Experimental results show that even state-of-the-art (SOTA) models, such as Qwen2.5-VL-72B, achieve an average performance of 48.8% on tasks involving abstract concepts. Additionally, performance drops by 15% as task complexity increases, highlighting the challenges LVLMs face in maintaining consistent performance. Through this work, we aim to reveal the limitations of current LVLMs and offer insights into how they can more effectively emulate human cognitive processes in the future.

## 1 INTRODUCTION

The rapid development of artificial intelligence (AI) has driven significant progress in LVLMs, enhancing their ability to process and interpret video data (Li et al., 2023; Zhang et al., 2023a; Lin et al., 2023; Li et al., 2024a; Ye et al., 2024; Tang et al., 2023). However, it remains unclear whether these models can truly emulate human-level general intelligence and cognitive abilities, such as symbolic understanding, abstract reasoning, and generalization (Tian et al., 2017; Hagendorff et al., 2023). While recent benchmarks for large language and vision models have begun incorporating cognition-oriented evaluations (Song et al., 2024b; Coda-Forno et al., 2024; Chia et al., 2024), existing benchmarks for LVLMs (Yu et al., 2019; Ning et al., 2023; Chen et al., 2023; Fang et al., 2024; Li et al., 2024e; Fu et al., 2024; Li et al., 2024b) focus mainly on semantic understanding, relying on web-crawled data that lack controllability over content and video difficulty, limiting their scalability as diagnostic tools. As a result, symbolic understanding and abstract reasoning are often evaluated implicitly, without directly testing core cognitive abilities. Additionally, cognitive science (Stillings, 1995) shows that games are used to study human cognition, offering a useful parallel for exploring how LVLMs might develop and demonstrate cognitive abilities. We aim to investigate how LVLMs perceive and interpret video content, generalise from symbolic and abstract elements about object properties such as size, colour, and shape, as well as dynamic attributes like motion type and speed, and higher-level spatial and temporal relationships.

To achieve it, we propose VideoCogQA, a controllable and scalable benchmark designed to assess the cognitive capabilities of LVLMs rigorously. VideoCogQA employs a fully programmatic video synthesis framework that offers fine-grained control over video content and task difficulty. Inspired by classic and widely recognized games such as maze navigation, sky battles, and others,

| Benchmarks | Understanding | Reasoning | Synthesis | Control | Difficulty Level |
|---|:---:|:---:|:---:|:---:|:---:|
| Video-Bench (Ning et al., 2023) | ✓ | ✗ | ✗ | ✗ | ✗ |
| MMBench-Video (Fang et al., 2024) | ✓ | ✗ | ✗ | ✗ | ✗ |
| AutoEval-Video (Chen et al., 2023) | ✓ | ✓ | ✗ | ✗ | ✗ |
| MVBench (Li et al., 2024b) | ✓ | ✓ | ✗ | ✗ | ✗ |
| Video-MME (Fu et al., 2024) | ✓ | ✓ | ✗ | ✗ | ✗ |
| VideoVista (Li et al., 2024e) | ✓ | ✓ | ✗ | ✗ | ✗ |
| VideoNIAH (Zhao et al., 2024) | ✓ | ✗ | ✓ | ✓ | ✗ |
| VideoCogQA (ours) | ✓ | ✓ | ✓ | ✓ | ✓ |

Table 1: Comparison of video benchmarks across key tasks and characteristics: understanding, reasoning, synthesis, controllability, and difficulty level.

we design a series of scenarios aimed at evaluating key cognitive dimensions in LVLMs, including Object Perception (Spelke, 1990), Action Perception (Kelso et al., 2018), Spatial Reasoning (Malik & Binford, 1983; Stock, 1998), Temporal Reasoning (Mark, 2020), and understanding within Gaming and Full-modal environments (Oei & Patterson, 2013; Spence & Feng, 2010; Cohn, 2016). One key advantage of synthetic video-based evaluation is its ability to precisely assess core cognitive abilities across modalities, without relying on prior semantic knowledge. For example, the model's ability to perceive actions can be evaluated on interpreting ball motions (e.g., bouncing, rotating, horizontal movement) where the balls differ in color, shape, and size, without relying on contextual semantic cues (e.g. a kitchen scene that implies cooking). Table 1 presents a comparative analysis of VideoCogQA and existing benchmarks. Through our evaluation of popular LVLMs, we observe that while many models perform well on simple video tasks, their capabilities degrade notably as task complexity increases. For instance, Qwen2vl-72B shows a 5.5% performance drop when additional objects are introduced in the Action Arena scene, followed by a 9.5% decline at the highest difficulty level. Moreover, VideoCogQA offers fine-grained evaluation, showing that models with similar performance on simple tasks diverge as task difficulty increases. Further analysis suggests that the performance drop in temporal tasks stems from the visual encoder's insufficient ability to grasp high-level abstract and symbolic concepts. We also show that models' performance on VideoCogQA is strongly correlated with real-world videoQA benchmarks, both at the benchmark and ability levels, and that training on VideoCogQA provides modest but consistent improvements on natural-video datasets. These findings underscore the inherent limitations of current models in video-based cognitive tasks and highlight the need for improved robustness in LVLMs. Hence, our main contributions are as follows:

- We introduce a Python-based video synthesis pipeline that enables the cost-effective generation of video content for cognitive capability testing. Adjustable parameters allow for controlling video difficulty, while GPT-4-generated QA templates are integrated with Python-based video generation and logging to create batched QA pairs.

- We introduce **VideoCogQA**, a scalable and fully controllable benchmark designed to rigorously assess the cognitive abilities of LVLMs across various tasks and cognitive dimensions inspired by video games.

- Our experiments demonstrate that even advanced LVLMs struggle with symbolic and abstract videos, particularly in difficult scenes, highlighting the need for models with stronger capabilities to handle such complex challenges.

## 2 RELATED WORK

### 2.1 VIDEO-LMMS AND BENCHMARK

Recent advancements in large multi-modal models (Zhang et al., 2023b; Liu et al., 2024b;a; Wang et al., 2024a) have greatly enhanced understanding and reasoning capabilities across various domains, especially in image-based tasks (Wu et al., 2023a; Fu et al., 2023; Zhang et al., 2024c; Tong et al., 2025). As multi-modal research continues to evolve, there is a growing shift from static images to dynamic temporal video (Li et al., 2024d). Early investigations in video understanding for LMMs, employing visual encoders, have shown promising results (Li et al., 2023; Zhang et al.,

Figure 1: In **Sky Battle** scene, symbolic icons are used to represent players, bullets, and enemies. Difficulty is controlled by varying number and speed of enemies. Questions such as "How many enemies are destroyed by player?" test the counting ability in game scenes.

2023a; Lin et al., 2023; Xu et al., 2023; Li et al., 2024d; Song et al., 2024a; Li et al., 2024a; Ye et al., 2024). Meanwhile, active research has focused on constructing benchmarks to assess LVLM capabilities (Ning et al., 2023; Chen et al., 2023; Fang et al., 2024; Li et al., 2024e; Fu et al., 2024; Li et al., 2024b). For instance, MVBench (Li et al., 2024b) provides a suite of task-specific videos covering various tasks, marking substantial progress in video comprehension. MMBench-Video (Fang et al., 2024) uses extended videos from YouTube and applies free-form questioning to simulate real-world video understanding tasks. However, most existing video-based benchmarks focus on human behavior and contextual understanding, often neglecting abstract cognitive tasks. In MVBench (Li et al., 2024b), for instance, when evaluating models' action perception abilities, prior knowledge from the video, such as a playground scene, making it easier to infer the action of running, can lead to shortcut learning. In contrast, our setting uses abstract objects that perform actions such as bouncing or rotating, requiring true motion perception. Moreover, the limited scalability of these benchmarks restricts their broader applicability. To address these limitations, we introduce VideoCogQA, a scalable and controllable dataset to assess a range of cognitive abilities.

## 2.2 SYNTHETIC DATASET

Synthetic datasets are cost-effective and avoid the practical challenges of manual annotation (Grauman et al., 2022; Chen et al., 2023), extensive prompt engineering (Li et al., 2024b), and risks of data leakage from pre-trained video corpora (Xu et al., 2024). Furthermore, synthetic benchmarks offer a controlled and scalable approach to the evaluation of AI models (Peng et al., 2024; Zhao et al., 2024). In language model evaluation, synthetic data (Maheshwari et al., 2024) has been used to create structured benchmarks. Similarly, in visual-language model research, synthetic images have been used in controlled experiments to systematically evaluate visual reasoning (Johnson et al., 2017; Hudson & Manning, 2019; Peng et al., 2024). Notably, the Abstraction and Reasoning Corpus (ARC) (Chollet, 2019) utilizes programmatically generated images to assess artificial general intelligence. In the video domain, early studies such as Cater (Girdhar & Ramanan, 2019) and Clevrer (Yi et al., 2019) leveraged 3D rendering engines (Blender, 2018) to generate synthetic videos. More recently, synthetic video-based benchmarks have evolved to incorporate multimodal elements. For example, VideoNIAH (Zhao et al., 2024) integrates textual and visual components into videos to evaluate comprehension models' retrieval, order, and counting ability. Compared to prior synthetic video benchmarks such as VideoNIAH (Zhao et al., 2024), which primarily target retrieval and order/counting in text–video composites, VideoCogQA focuses on a balanced coverage of six cognitive abilities with explicit difficulty control and game-inspired, fully programmatic scenes. We adopt a Python-based video synthesis framework to construct a scalable and controllable dataset for the evaluation of LVLM. Its synthetic nature enables in-depth analysis of understanding and reasoning abilities beyond existing datasets.

## 3 VIDEOCOGQA

### 3.1 DATASET DESIGN

Recent advancements in language models have increasingly focused on evaluating their cognitive abilities in the context of video understanding (Li et al., 2024e). Common evaluation dimensions

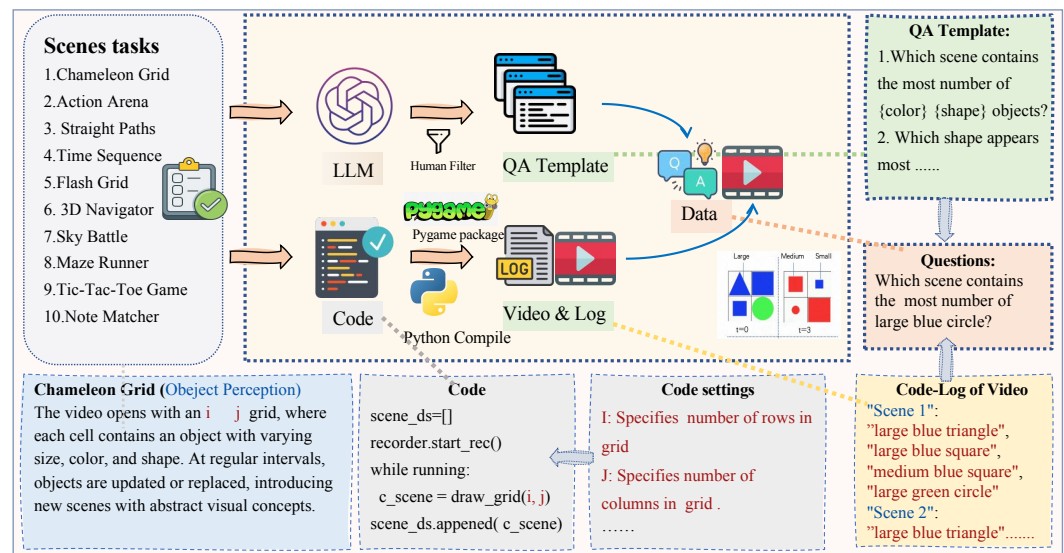

Figure 2: Pipeline for generating videos and corresponding QA templates: The variables $i$ and $j$ control the complexity of video scenes. A Python program executes and logs these variables, while GPT-4 generates scene-related question templates, which are then refined through human filtering. Finally, the generated code, QA pairs, and videos are collected. Cases are detailed in Appendix A.1.

include Object Perception (OP), Action Perception (AP), Temporal Reasoning (TR), and Spatial Reasoning (SR). In VideoCogQA, we expand this scope of video cognition assessment by introducing two key dimensions: Game-environment Perception (GP) and Full-modal Perception (FP). The synthesized video scenes incorporate symbolic elements and abstract concepts, containing symbolic objects, abstract attributes (color, shape, and size), abstract actions (action type, action speed, and direction), and spatial (2D and 3D scenes) and temporal relationships at varying levels of task difficulty. The following section provides specific descriptions of each dimension.

- **Object Perception (OP)**: This dimension involves precise recognition of symbolic objects varying in color, shape, and size (Wang et al., 2025a). It requires models to sustain high recognition accuracy across diverse visual and abstract attributes.

- **Action Perception (AP)**: This task assesses the model's proficiency in interpreting actions performed by symbolic objects (Chen et al., 2024), accounting for variations in action speed, direction, and type.

- **Temporal Reasoning (TR)**: This dimension focuses on the model's ability to understand and reason through sequences of events in videos (Chu et al., 2023; Fatemi et al., 2024; Cai et al., 2024), challenging it to track and interpret temporal relationships effectively.

- **Spatial Reasoning (SR)**: This task measures the model's understanding of spatial relationships within both 2D and 3D contexts (Wu et al., 2024b; Tang & Kejriwal, 2024), addressing abstract elements such as object positioning, orientation, and relative location within video content.

- **Game-environment Perception (GP)**: This dimension evaluates the model's ability to understand and reason about simulated game environments, which involve abstract concepts (Wu et al., 2023b; Topsakal & Harper, 2024). It focuses on the model's ability to interpret game mechanics, predict player actions, and understand the overall game structure, which is critical for LVLMs in game-related video analysis.

- **Full-modal Perception (FP)**: This dimension assesses the model's ability to integrate and process information across multiple modalities—visual, textual, and auditory (Li et al., 2024c). This cross-modal interaction involving symbolic objects is essential for advanced applications in video analysis.

> 1. How many enemies are destroyed by the player's plane?
> 2. At what timestamp does the player defeat the first enemy?
> 3. What is the maximum number of enemies visible on screen at any time?  ......

Figure 3: Automatically Generated Questions by GPT-4 in the Sky Battle Scene.

## 3.2 AUTOMATED VIDEO AND QA GENERATION

Guided by formal definitions of video cognitive abilities, we developed a synthetic video generation pipeline using Python, inspired by video game environments. This pipeline renders task-specific scenes incorporating symbolic elements and abstract concepts, while built-in randomness ensures variability in each generated video. Videos are produced in batches, and both temporal and spatial complexity are precisely controlled via code parameters. Scene logging, combined with paired question templates, allows for targeted evaluation of cognitive abilities. The following sections provide detailed descriptions of each video scene used in **VideoCogQA**.

1. **Chameleon Grid (OP-S1)**: A grid of size $i \times j$ where each cell contains a symbolic object with attributes such as size (small, medium, large), color (red, green, blue), and shape (triangle, circle, square). Objects are periodically updated, creating dynamic visual stimuli inspired by *Bejeweled* and *Candy Crush*. The complexity is controlled by the grid dimensions. This scene evaluates the model's ability to recognize and count objects under changing spatial arrangements. Example question: **How many red circles appear across all scenes?**

2. **Action Arena (AP-S2)**: A scene with $n$ objects performing $a$ action types, such as horizontal movement, jumping, scaling, or rotation. Complexity is adjusted by varying object count and action diversity. The task assesses the model's ability to distinguish and classify action types in a dynamic environment. Example question: **What action is the red ball performing?**

3. **Straight Paths (AP-S3)**: $n$ symbolic objects move linearly, bouncing off walls and changing direction, each with one of $a$ speed types. Complexity is varied through object count and speed range. This task is designed to measure the model's ability to estimate speed, track direction, and predict future positions. Example question: **Which object moves faster, the green circle or the blue circle?**

4. **Time Sequence (TR-S4)**: Symbolic objects appear and disappear at fixed intervals with a simulated clock display, inspired by *Simon* and *Guitar Hero*. Complexity is controlled by the number of objects $n$ and interval length $t$. This scene tests the model's ability to track the temporal order and sequence of events. Example question: **Which object appears immediately after the red circle?**

5. **Flash Grid (SR-S5)**: A $i \times j$ matrix where objects randomly appear in different cells, inspired by *Memory Matrix* and *Whac-A-Mole*. Complexity is determined by grid size. This task evaluates the model's ability to memorize transient spatial positions and reason about spatial relations. Example question: **For the second occurrence of the object, what is in the cell to its right?**

6. **3D Navigator (SR-S6)**: A 3D environment with symbolic objects (e.g., cubes, pyramids), where a small ball moves randomly along edges, inspired by *Super Monkey Ball*. Complexity is adjusted by speed $t$ and path complexity $e$. This task measures the model's ability to track motion and predict trajectories in 3D space. Example question: **Which edge of the cube is the red ball traversed most frequently?**

7. **Sky Battle (GP-S7)**: A player-controlled plane interacts with enemy icons and bullets, inspired by classic arcade shooters. Complexity is varied by the number $n$ and speed $a$ of enemies. The task evaluates the model's ability to perceive symbolic gameplay environments and interpret game mechanics. Example question: **How many enemy planes were destroyed by the player?**

8. **Maze Runner (GP-S8)**: A symbolic object navigates an $i \times j$ maze toward a goal, inspired by puzzle games. Complexity is adjusted via maze design. The task examines the model's

ability to plan navigation and evaluate symbolic environments. Example question: **How many steps did the player take to solve the maze?**

9. **Tic-Tac-Toe Game (GP-S9)**: A simulated tic-tac-toe game on a $3 \times 3$ grid. The task assesses the model's ability to perceive gameplay rules and reason about symbolic outcomes. Example question: **How many moves did each player make?**

10. **Note Matcher (FP-S10)**: A symbolic object is paired with musical notes (1–7), inspired by *Patapon*. Complexity is varied by note count $n$ and frequency of object changes $t$. This scene tests the model's ability to link sounds with visuals and combine multiple modalities. Example question: **Which musical note is associated with the appearance of the ball?**

| Scene | Parameters | Easy | Medium | Hard |
|---|---|---|---|---|
| **Chameleon Grid** | I: Number of cells per row | I=2 | I=5 | I=8 |
| | J: Number of cells per column | J=2 | J=5 | J=8 |

Table 2: Parameters of difficulty level in Chameleon Grid. All scenes are in Appendix A.2.

We generate videos for predefined scenes using Python, enabling fine-grained control over task difficulty via code parameters, as shown in Figure 2. Each video lasts about 30 seconds. By repeatedly executing the code with built-in randomness (such as variations in the object's color, size, and shape), we efficiently produce large batches of videos, ensuring scalability. To construct QA pairs, we prompt GPT-4 with: "The above is the code to generate a game video using Pygame. Provide a series of QA templates related to it." An example from *Sky Battle* is shown in Figure 3. (see Appendix A.2 for the full prompts). For evaluation, we adopt multiple-choice questions with 3-5 shuffled options. In total, **VideoCogQA** includes 800 videos and 3,280 questions across ten Python-based game scenarios. Each scene contains approximately 300 questions, with about 100 questions per difficulty level. Each template consists of roughly 10 questions per difficulty. VideoCogQA focuses on a balanced coverage of six core cognitive abilities under fully controllable synthetic scenes. This balanced yet controllable design makes it possible to attribute model failures to specific cognitive skills, instead of confounding factors such as dataset bias or annotation noise shown in Table 3. We show more cases, along with additional potential capability tests through Python-generated videos in Appendix A.1.

## 4 EXPERIMENT

### 4.1 SETUP

We evaluate widely used open-source Large Video-Language Models (LVLMs) that have been fine-tuned on videoQA pairs. These models include MiniCPM-V (Yao et al., 2024), Video-LLaMA2 (Boqiang Zhang, 2025), InternVideo2 (Wang et al., 2025b), Video-LLaVA (Lin et al., 2023), Video-R1 (Feng et al., 2025), LongVILA-R1-7B (Chen et al., 2025), LLaVA-OneVision (Li et al.), LLaVA-NEXT-Video-34B/7B (Zhang et al., 2024b), and InternLM-XComposer-2.5 (Zhang et al., 2024a). By incorporating a diverse set of recent architectures, such as Video-LLaMA and Qwen-VL, our evaluation captures the latest advances in video-LLMs. Additionally, we include two recent large-scale models, Qwen2-VL-72B (Wang et al., 2024a) and Qwen2.5-VL-72B (Bai et al., 2025), along with proprietary LVLMs, Gemini-Flash-2.0 and GPT-4o. While InternVideo supports audio encoding, we adopt a standardized input format by using musical note representations, ensuring consistency and fairness across models. For fairness, each model is evaluated using its official default inference settings, including the number of frames processed. Additional model details can be found in Appendix A.3. Following prior work (Li et al., 2024b), we preface each model prompt with the prefix "`Best Option:`". Human performance is reported as the average accuracy of five independent annotators. Note that humans watch the full continuous videos, making this human upper bound more favorable than the 2 FPS, max-64-frame setting used for LVLMs. (Appendix A.4).

| Ability | VideoMME | | VideoCogQA | |
|---|---|---|---|---|
| | QAs | Videos | QAs (Easy / Med / Hard) | Videos |
| OP (Object) | 1000 | 693 | 328 (109 / 109 / 110) | 80 |
| AP (Action) | 600 | 432 | 656 (218 / 218 / 220) | 160 |
| TR (Temporal) | 232 | 220 | 328 (109 / 109 / 110) | 80 |
| SR (Spatial) | 110 | 104 | 656 (218 / 218 / 220) | 160 |
| GP (GameWorld) | N/A | N/A | 984 (327 / 327 / 330) | 240 |
| FP (Full-modal) | N/A | N/A | 328 (109 / 109 / 110) | 80 |
| Other (OCR / Info) | 758 | 559 | N/A | N/A |
| Total | 2700 | 2008 | 3280 | 800 |

Table 3: Ability-based distribution of questions and videos in VideoMME and VideoCogQA. VideoCogQA provides a balanced yet controllable coverage of six cognitive abilities with explicit difficulty levels.

| Model | OP | AP | | TR | SR | | GP | | | FP | Avg. |
|---|---|---|---|---|---|---|---|---|---|---|---|
| | S1 | S2 | S3 | S4 | S5 | S6 | S7 | S8 | S9 | S10 | |
| **Open-Source Models** | | | | | | | | | | | |
| MiniCPM-V | 28.2 | 49.5 | 39.3 | 47.8 | 32.2 | 34.7 | 28.9 | 26.7 | 46.0 | 54.4 | 38.8 |
| Video-LLaMA2 | 31.3 | 50.5 | 33.5 | 48.3 | 36.4 | 18.7 | 26.7 | 27.8 | 52.0 | 52.2 | 37.7 |
| InternVideo2 | 35.3 | 36.5 | 32.5 | 43.3 | 33.4 | 21.7 | 23.4 | 23.8 | 46.0 | 51.5 | 34.7 |
| Video-LLaVA | 40.4 | 21.0 | 40.8 | 23.2 | 37.5 | 21.3 | 16.7 | 25.5 | 38.0 | 60.0 | 32.4 |
| Video-R1-7B | 46.0 | 52.0 | 39.3 | 60.8 | 55.8 | 54.7 | 46.7 | 36.7 | 56.7 | 72.2 | 52.1 |
| LongVILA-R1-7B | 33.6 | 38.8 | 42.3 | 42.7 | 34.7 | 34.7 | 32.2 | 30.0 | 39.3 | 41.1 | 36.9 |
| LLaVA-OneVision-7B | 42.0 | 49.2 | 41.8 | 51.2 | 46.4 | 48.0 | 46.3 | 37.8 | 49.3 | 56.7 | 46.9 |
| LLaVA-NEXT-Video-7B | 20.4 | 22.5 | 30.7 | 21.0 | 33.8 | 18.7 | 12.2 | 14.4 | 15.3 | 46.7 | 23.6 |
| LLaVA-NEXT-Video-34B | 28.4 | 42.0 | 42.7 | 39.0 | 22.9 | 58.0 | 37.8 | 12.2 | 33.3 | 58.9 | 37.5 |
| InternLM-XComposer-2.5 | 36.0 | 38.2 | 45.5 | 44.5 | 43.1 | 20.0 | 8.9 | 25.6 | 35.3 | 61.1 | 35.8 |
| Qwen2-VL-72B | **51.8** | 58.2 | 60.0 | 56.8 | 60.7 | 42.0 | 32.2 | 37.8 | 62.0 | 76.7 | 53.8 |
| Qwen2.5-VL-72B | 48.8 | 57.2 | **61.0** | 56.8 | 61.7 | 42.0 | 34.2 | 38.8 | 62.5 | **78.2** | 54.1 |
| **Closed-Source Models** | | | | | | | | | | | |
| Gemini-Flash /w 32 frames | 41.2 | 45.2 | 41.8 | **68.5** | 50.0 | 37.3 | 27.8 | 40.0 | 55.5 | 71.1 | 47.8 |
| GPT-4o /w 32 frames | 44.2 | **64.8** | 51.5 | 61.3 | **67.3** | **68.7** | **60.0** | **52.2** | **72.0** | 70.0 | **66.1** |
| Average models | 37.7 | 44.7 | 43.1 | 46.5 | 44.0 | 37.2 | 31.0 | 30.7 | 47.4 | 60.8 | 42.3 |
| Human | 88.9 | 91.3 | 89.7 | 92.1 | 87.5 | 85.2 | 90.4 | 93.6 | 95.1 | 89.0 | 90.3 |

Table 4: Performance of LVLMs across different scenes.

## 4.2 MAIN RESULTS

VideoCogQA reveals LVLMs' capabilities, their cognitive gap from humans, and the promising performance of reasoning models. As shown in Table 4, most models struggle with Object Perception (OP), Spatial Reasoning (SR), and Game-environment Perception (GP) tasks, which require advanced handling of visual abstract concepts. In contrast, Action Perception (AP) and Temporal Reasoning (TR) primarily focus on abstract action recognition and the temporal reasoning of objects, with fewer symbolic elements, where LVLMs perform relatively. Their strong performance in Full-modal Perception (FP) reflects a solid grasp of integrated audio-visual information, particularly when converting musical notes to text. The advanced Qwen2.5-VL-72B model stands out among the open-source models tested, consistently achieving the highest accuracy across most tasks, including OP (48.8%), AP (59.1%), TR (56.8%), SR (51.8%), GP (45.2%), and FP (78.2%), leading to an impressive overall average accuracy of 54.1%. Compared to other models, MiniCPM-V shows competitive results in AP (44.4%) and TR (47.8%), while LLaVA-NEXT-Video-34B excelled in SR

| Model | Diff. | OP | AP | | TR | SR | | GP | | | FP | Avg. |
|---|---|---|---|---|---|---|---|---|---|---|---|---|
| | | S1 | S2 | S3 | S4 | S5 | S6 | S7 | S8 | S9 | S10 | |
| MiniCPM-V | Easy | 34.7 | 54.5 | 50.5 | 53.5 | 47.3 | 36.0 | 43.3 | 30.0 | 46.0 | 70.0 | 46.3 |
| | Medium | 26.0 | 48.5 | 36.5 | 50.5 | 30.0 | 40.0 | 23.3 | 26.7 | — | 43.3 | 35.8 |
| | Hard | 24.0 | 45.5 | 31.0 | 39.5 | 19.3 | 28.0 | 20.0 | 23.3 | — | 50.0 | 31.0 |
| Video-LLaMA2 | Easy | 41.3 | 61.0 | 46.5 | 50.0 | 49.3 | 20.0 | 26.7 | 33.3 | 52.0 | 70.0 | 44.8 |
| | Medium | 29.3 | 52.0 | 26.5 | 53.0 | 31.3 | 18.0 | 23.3 | 31.0 | — | 56.7 | 34.2 |
| | Hard | 23.3 | 38.5 | 27.5 | 42.0 | 28.7 | 18.0 | 30.0 | 30.0 | — | 30.0 | 29.6 |
| LLaVA-NEXT-Video-34B | Easy | 29.3 | 48.5 | 53.5 | 44.0 | 42.0 | 62.0 | 56.7 | 21.0 | 33.3 | 70.0 | 44.7 |
| | Medium | 26.0 | 40.0 | 42.5 | 45.0 | 18.7 | 54.0 | 36.7 | 16.7 | — | 56.7 | 36.9 |
| | Hard | 30.0 | 37.5 | 32.0 | 28.0 | 8.0 | 58.0 | 20.0 | 20.0 | — | 50.0 | 31.4 |
| Qwen2-VL-72B | Easy | 61.3 | 65.0 | 67.0 | 64.5 | 76.0 | 52.0 | 40.0 | 63.3 | 62.0 | 83.3 | 63.3 |
| | Medium | 48.7 | 59.5 | 62.5 | 56.0 | 56.0 | 34.0 | 33.3 | 23.3 | — | 80.0 | 50.1 |
| | Hard | 45.3 | 50.0 | 50.5 | 50.0 | 50.0 | 40.0 | 23.3 | 26.7 | — | 66.7 | 44.4 |

Table 5: Performance across scenes and difficulty levels shows poor robustness at higher difficulties.

| Model | 8 frames | 16 frames | 32 frames | 64 frames | 128 frames |
|---|---|---|---|---|---|
| Qwen2.5-VL-3B | 46.3 | 46.8 | 47.9 | 47.3 | 44.2 |
| Qwen2.5-VL-7B | 50.9 | 53.0 | 54.9 | 55.1 | 54.0 |
| Qwen2.5-VL-72B | 54.9 | 56.5 | 57.1 | 57.0 | 57.8 |

Table 6: Frame sampling analysis on VideoCogQA. Accuracy of Qwen2.5-VL models when varying the number of uniformly sampled frames per video from 8 to 128.

(40.4%). Notably, Video-R1 achieves a strong performance of 52.1 with the 7B model, demonstrating its ability to output the thinking process, while LongVILA-R1, designed for long video understanding, shows inferior performance. Despite these achievements, even the top models still fall far short of human-level performance (90.3%).

## 4.3 PERFORMANCE ACROSS DIFFICULTY LEVELS

Different task difficulties in VideoCogQA expose the limited generalization of current models and enable more effective fine-grained evaluation. Table 5 and Figure 6 illustrate a detailed evaluation of model performance across varying difficulty levels. As task difficulty increases, all models show a consistent decline in accuracy, emphasizing the inherent challenges in complex video cognition tasks. Specifically, most models experience a drop of approximately 10 percentage points from Easy to Medium difficulty, followed by an additional 5-point decline at the Difficult level. Notably, while Video-LLaMA2 and LLaVA-NEXT-Video-34B show similar performance at the Easy level, LLaVA-NEXT-34B begins to outperform Video-LLaMA2 as tasks become more challenging. Further results are provided in Appendix A.8.

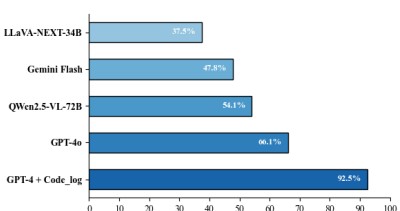

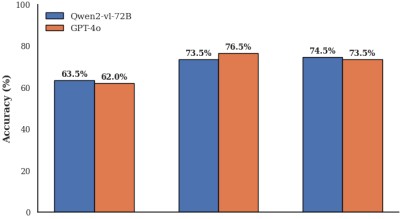

Figure 4: Comparison of VideoLLM performance, where GPT-4 is evaluated using video text descriptions derived from code logs.

Figure 5: Model Performance on Object Size, Color, and Shape on the sing-frame perception tasks from scene1.

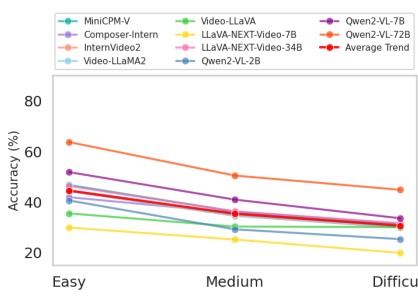 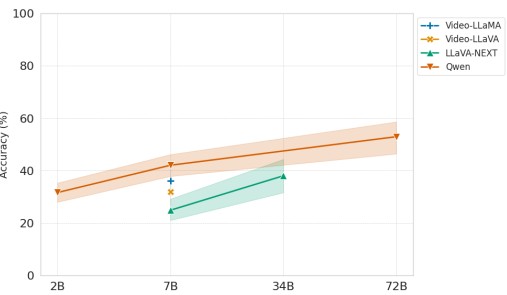

Figure 6: Performance of LVLMs across different levels.

Figure 7: Performance of LVLMs under different model parameters.

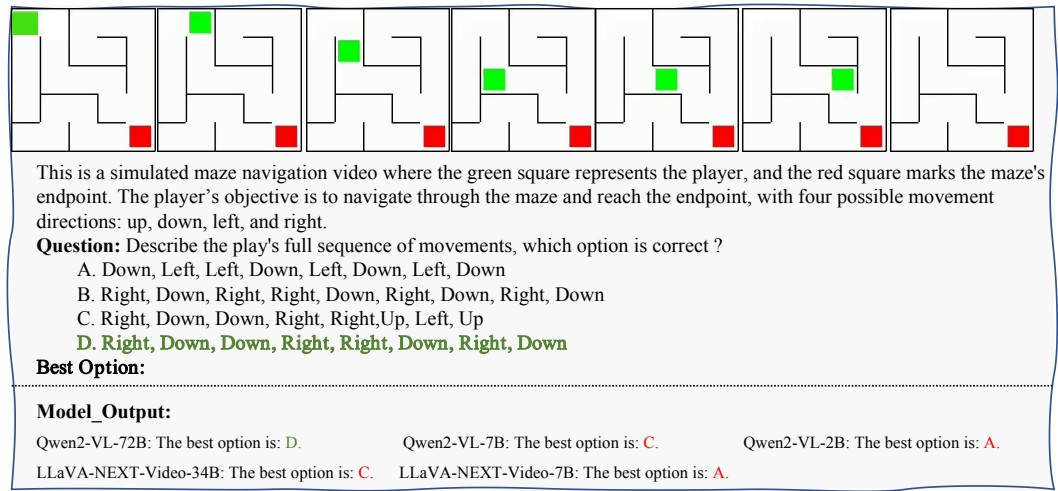

Figure 8: Maze Runner cases. The task requires describing the trajectory of the green square player navigating the maze to reach the red square goal. Only Qwen2VL-72B produced the correct answer.

## 4.4 INSIGHTS INTO VIDEOLLM PERFORMANCE

Our analysis suggests that LVLMs' struggles with temporal tasks are largely due to limitations in visual encoders, rather than solely to the LLM's reasoning capabilities. When raw videos are substituted with symbolic descriptions from code logs, models such as GPT-4 show clear improvements in accuracy (Figure 4). This indicates that the main bottleneck lies in the models' ability to extract structured cues from the visual input. To further investigate symbolic perception, we design a diagnostic test using Chameleon Grid (S1) videos, shown in Figure 10. In each single frame, objects are arranged in an $i \times j$ grid. The model is given a single frame, with the prompt instructing it to identify and output each object's size, color, and shape sequentially from left to right, top to bottom. While trivial for humans, both models frequently misidentify object size (Figure 5). These results suggest that widely used visual encoders, such as CLIP (Radford et al., 2021), are not sufficiently trained in symbolic elements, particularly in distinguishing object sizes, emphasizing the need for better training approaches that could integrate more abstract concepts.

## 4.5 IMPACT OF MODEL SIZE ON PERFORMANCE

As shown in Figure 7, there is a strong positive correlation between model size and performance. For instance, scaling the Qwenvl model from 2B to 7B and then to 72B parameters leads to substantial improvements in average performance, increasing from 31.9% to 42.5% and ultimately to 53.7%, respectively.

| Dataset | 8 Frames | 16 Frames | 32 Frames | 64 Frames | Corr. Coeff. |
|---|---|---|---|---|---|
| VideoCogQA | 50.9 | 53.0 | 54.9 | 55.1 | 1.0000 |
| LongVideoBench | 52.6 | 56.8 | 58.5 | 59.0 | 0.9836 |
| VideoMME | 53.6 | 58.8 | 60.6 | 63.2 | 0.9775 |
| LVBench | 33.7 | 36.5 | 39.6 | 40.0 | 0.9984 |

Table 7: Performance across benchmarks with different frame sampling. The last column reports correlation coefficients with VideoCogQA.

| Models | OP | | AP | | TR | | SR | | GP | | FP | |
|---|---|---|---|---|---|---|---|---|---|---|---|---|
| | Origin | New | Origin | New | Origin | New | Origin | New | Origin | New | Origin | New |
| Qwen2.5-VL-3B | 46.8 | 47.5 | 42.5 | 41.0 | 50.2 | 51.0 | 45.2 | 46.5 | 42.3 | 41.5 | 52.2 | 52.5 |
| Qwen2.5-VL-7B | 45.6 | 46.5 | 48.3 | 49.0 | 56.1 | 57.5 | 50.2 | 50.5 | 44.4 | 42.5 | 57.2 | 57.0 |
| Qwen2.5-VL-72B | 51.8 | 52.5 | 59.1 | 61.0 | 56.8 | 56.0 | 51.3 | 50.0 | 44.8 | 43.5 | 76.7 | 75.5 |
| **Corr. Coeff.** | 0.9996 | | 0.9984 | | 0.9496 | | 0.9597 | | 0.9308 | | 0.9996 | |

Table 8: Performance of models across various tasks (OP, AP, TR, SR, GP, FP). "Origin" denotes the original dataset, and "New" represents the new sample size setting.

### 4.6 CASE STUDY

Figure 8 presents a simple case from the Maze Runner scene, where the LVLM must track the movement trajectory of a green block. This task, which is trivial for humans, requires the model to accurately perceive and retain the path structure. Among all models, only Qwen2-VL-72B correctly selected option D, demonstrating a strong understanding of the game environment and the optimal path. In contrast, Qwen2-VL-7B and LLaVA-NEXT-Video-34B chose option C, indicating partial understanding of the spatial reasoning task. Interestingly, the performance of Qwen2-VL-2B was comparable to that of the larger LLaVA-NEXT-Video-7B, highlighting the efficiency of the Qwen series in handling complex video tasks despite its smaller size.

### 4.7 BENCHMARK EVALUATION AND ANALYSIS

To validate the effectiveness of VideoCogQA, we compare its results with real-world video benchmarks, including VideoMME (Fu et al., 2024), LongVideoBench (Wu et al., 2024a), and LVBench (Wang et al., 2024b). We also analyze the impact of benchmark size by randomly generating objects and re-synthesizing 20 videos per scene, each with 100 questions, resulting in 200 videos and 1,000 questions. Three models from the Qwen2.5-VL models are evaluated in this setting. As the number of frame samples increases, performance improves consistently. Importantly, all correlation coefficients exceed 0.9, confirming its reliability as a proxy for real-world video understanding (Table 7). Furthermore, results from VideoCogQA remain highly correlated with the original datasets (Table 8), demonstrating both its robustness to sample size and its relevance for evaluating real-world video cognition. We show that performance on VideoCogQA is strongly correlated with natural-video QA benchmarks at both the benchmark and ability levels (More results are provided in Appendix A.6), and we further examine whether training on the synthetic tasks in VideoCogQA transfers to natural-videoQA (see Appendix A.7).

## 5 CONCLUSION

In this work, we introduce VideoCogQA, a scalable and controllable benchmark for assessing the cognitive abilities of Large Video-Language Models (LVLMs). Unlike previous benchmarks based on real-world videos with limited controllability, VideoCogQA is programmatically designed to allow precise control over visual content and task difficulty, facilitating fine-grained evaluation of symbolic and abstract reasoning. Experimental results show that even state-of-the-art models, such as GPT-4o and Qwen2.5-VL-72B, struggle with symbolic reasoning tasks, with performance declining significantly as complexity increases. These findings highlight the need for improved generalization of cognitive abilities in LVLMs.

## ETHICS STATEMENT

This work adheres to the ICLR Code of Ethics. The datasets utilized in this work are sourced from Python-based automated synthesis of game scenes, which feature symbolized abstract elements. These datasets are safe for use and do not present any ethical concerns. We have taken care to avoid any biases or discriminatory outcomes in our research process. No personally identifiable information was used, and no experiments were conducted that could raise privacy or security concerns. We are committed to maintaining transparency and integrity throughout the research process.

## REPRODUCIBILITY STATEMENT

We have made every effort to ensure the reproducibility of the results presented in this paper. We introduce a pipeline for dataset generation in Section 3, along with the specific prompts provided in Appendix A.2, which outlines the construction of the benchmark. For evaluation, the models and their configurations are detailed in Section 4.1 and Appendix A.3. We believe these measures will enable other researchers to reproduce our work and further advance the field.

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

> {} The above is the code to generate a game video using Pygame. The QA template is: {} We want to test {} ability in this scene. Please refine and filter about 10 non-redundant QA templates.

Figure 9: The prompt used to refine QA templates.

## A APPENDIX

### A.1 ADDITIONAL CASES

We present more examples of scenes in Figures 10, 11, 12, 13, 14, 15, 16, 17, and 19. These figures highlight the diversity and complexity of the scenes in our benchmark, offering visual insights into tasks that evaluate various cognitive abilities. Beyond these, additional videos can be synthesized using Python to further assess VideoLLM's capabilities in different tasks. More cases are shown in Figures 20, 21, and 22. These scenarios can also control difficulty by adjusting the number of objects, testing abilities such as pattern recognition, object tracking, and fine-grained visual perception.

### A.2 DETAILED PARAMETERS AND QA TEMPLATES

Table 9 summarizes the parameter settings for the complex scenes, and Table 31 lists the QA templates used for each scene. These templates span diverse reasoning tasks, including temporal, spatial, and action-based queries.

To construct QA templates in a scalable yet controllable way, we use a fixed prompt with scene-specific inputs. For each scene, we provide (i) the full Python/Pygame code, (ii) a set of seed QA templates, and (iii) the target cognitive ability. The core user prompt (Figure 9) contains three {} slots, which are filled with these three components respectively. We consider abilities such as *object perception*, *action perception*, *temporal reasoning*, and *spatial reasoning*. GPT-4 is instructed to stay grounded in code-defined objects and events, avoid semantic redundancy, and remain within the specified ability.

### A.3 MODELS

We evaluate large video language models (LVLMs) covering distinct types:

- **MiniCPM-V** (Yao et al., 2024): A vision-language MLLM with 8B parameters. MiniCPM-V 2.6 outperforms GPT-4V in single-image, multi-image, and video understanding tasks. It surpasses GPT-4o mini, Gemini 1.5 Pro, and Claude 3.5 Sonnet in single-image understanding. It also offers enhanced OCR capabilities, trustworthy behavior, multilingual support, and real-time video understanding on devices like the iPad.

- **Video-LLaMA2** (Cheng et al., 2024): A 7B parameter model with a Spatial-Temporal Convolution (STC) connector, designed to capture the spatial and temporal dynamics in video data.

- **InternVideo2** (Wang et al., 2024c): A unified model combining masked video modeling, video-audio-text contrastive learning, and next-token prediction. The 7B parameter model excels in video dialogue and long video understanding tasks. It is used with a video BLIP module and the Mistral-7B LLM for evaluation, where the video encoder is updated during training.

- **Video-LLaVA** (Lin et al., 2023): A 7B parameter model trained on a mixed dataset of images and videos, designed to bind visual signals to the language feature space. It extracts features from video frames through language binding and offers unified visual representations.

| Scene | Parameters | Easy | Medium | Hard |
|---|---|---|---|---|
| **Chameleon Grid** | I: Number of cells per row
J: Number of cells per column | I=2
J=2 | I=5
J=5 | I=8
J=8 |
| **Action Arena** | N: Number of objects
A: Number of action types | N=3
A=3 | N=6
A=6 | N=9
A=8 |
| **Straight Paths** | N: Number of objects
A: Number of speed types | N=3
A=3 | N=6
A=5 | N=9
A=8 |
| **Time Sequence** | T: Time interval of changes
N: Number of objects | T=5
N=3 | T=3
N=5 | T=1
N=8 |
| **Flash Grid** | I: Number of cells per row
J: Number of cells per column | I=2
J=2 | I=5
J=5 | I=8
J=8 |
| **3D Navigator** | T: Time to traverse each edge
E: Number of edges | T=2
E=5 | T=1
E=8 | T=0.5
E=12 |
| **Sky Battle** | N: Number of enemy planes
A: Enemy speed types | N=3
A=2 | N=5
A=5 | N=10
A=8 |
| **Maze Runner** | I: Maze length
J: Maze width | I=3
J=3 | I=5
J=5 | I=8
J=8 |
| **Note Matcher** | T: Time interval of changes
N: Number of notes | T=5
N=3 | T=3
N=5 | T=1
N=7 |
| **Tic-Tac-Toe Game** | No difficulty levels | None | None | None |

Table 9: Detailed parameters for all scene types across difficulty levels.

- **Video-R1** (Feng et al., 2025): A 7B parameter model developed to enhance video representation and understanding by optimizing both spatial and temporal video features. Video-R1 incorporates dynamic video encoding to capture long-range dependencies and demonstrates strong performance on tasks such as video summarization, action recognition, and complex multimodal reasoning. Additionally, it utilizes the R1 paradigm, inspired by DeepSeek-R1's success in eliciting reasoning abilities via rule-based reinforcement learning (RL), making it a pioneering model for video reasoning in MLLMs.

- **LongVILA-R1-7B** (Chen et al., 2025): A 7B parameter model designed for video understanding, emphasizing the integration of long-duration temporal information. LongVILA-R1 excels in tasks like video question answering and long-term event prediction, leveraging a two-stage training pipeline with chain-of-thought supervised fine-tuning (CoT-SFT) and reinforcement learning (RL).

- **LLaVA-OneVision** (Li et al.): A 7B parameter multimodal model that bridges the gap between vision and language by utilizing one-shot vision-language pretraining. LLaVA-OneVision demonstrates exceptional performance across single-image, multi-image, and video tasks, advancing the capabilities of open large multimodal models (LMMs). Notably, it showcases strong transfer learning across different modalities and scenarios, allowing for effective task transfer from images to videos and driving new emerging capabilities in multimodal retrieval and cross-modal generation.

- **LLaVA-NEXT-Video** (Zhang et al., 2024b): A model incorporating a 4K-token context window and using CLIP ViT-L/14 for feature extraction. We evaluate both the 7B and 34B parameter versions.

- **InternLM-XComposer-2.5** (Zhang et al., 2024a): A 7B parameter model treating videos as ultra-high-resolution composite images. It excels in text-image comprehension and composition, achieving performance comparable to GPT-4V.

- **Qwen-VL** (Wang et al., 2024a): A model integrating Multimodal Rotary Position Embedding (M-RoPE) for effective fusion of positional information across text, images, and

videos. We evaluate the 2B, 7B, and 72B versions, with the 72B model matching GPT-4o and Claude 3.5 Sonnet in multimodal tasks.

| Dataset | VideoCogQA | LongVideoBench | VideoMME | LVBench |
|---|---|---|---|---|
| VideoCogQA | 1.0000 | 0.9836 | 0.9775 | 0.9984 |
| LongVideoBench | 0.9836 | 1.0000 | 0.9951 | 0.9902 |
| VideoMME | 0.9775 | 0.9951 | 1.0000 | 0.9857 |
| LVBench | 0.9984 | 0.9902 | 0.9857 | 1.0000 |

Table 10: Correlation coefficients across benchmarks.

### A.4 HUMAN BASELINE AND DIFFICULTY VALIDATION

To characterize the human ceiling and validate the difficulty design of VideoCogQA, we conduct a controlled human evaluation.

**Participants and evaluation structure.** We recruit five annotators (H1–H5), and each annotator completes the full evaluation twice with a 48-hour gap between sessions. The evaluation covers 10 scenes (S1–S10), each containing 20 videos and 2–3 questions per video (approximately 50 questions per scene). Each scene is further divided into three difficulty levels—Easy, Medium, and Hard—with roughly 6–7 videos per level. All human results are reported as averages over the two sessions.

**Viewing procedure.** Annotators may preview the questions before or during video viewing. Each video may be replayed at most three times. Importantly, annotators watch the full continuous video (i.e., without frame subsampling), and answers cannot be revised once submitted.

**Accuracy by scene and difficulty.** Table 11 reports per-scene accuracies for each annotator, along with per-participant averages across all scenes. Table 12 presents aggregated accuracy by difficulty level. Overall human accuracy on VideoCogQA is approximately $90.0\%$, confirming a high human ceiling and a clear separation between difficulty levels.

| | S1 | S2 | S3 | S4 | S5 | S6 | S7 | S8 | S9 | S10 | AVG |
|---|---|---|---|---|---|---|---|---|---|---|---|
| H1 | 91.2 | 93.1 | 88.4 | 90.7 | 87.3 | 86.5 | 92.8 | 96.1 | 97.0 | 90.4 | 91.3 |
| H2 | 88.6 | 94.0 | 92.4 | 94.8 | 89.6 | 83.4 | 88.2 | 92.4 | 96.6 | 88.6 | 90.2 |
| H3 | 90.2 | 89.4 | 87.6 | 95.2 | 90.2 | 82.6 | 91.6 | 91.2 | 93.8 | 87.4 | 89.9 |
| H4 | 92.2 | 88.6 | 93.4 | 90.2 | 85.4 | 87.2 | 89.6 | 97.0 | 97.8 | 92.4 | 91.4 |
| H5 | 86.8 | 90.6 | 92.0 | 94.4 | 88.2 | 88.6 | 89.0 | 95.0 | 94.4 | 90.2 | 90.9 |

Table 11: Human accuracies (%) by scene and annotator. The last column (AVG) is the per-annotator average over all scenes.

| Difficulty | Easy | Medium | Hard |
|---|---|---|---|
| Avg human accuracy | 98.2 | 91.6 | 81.1 |

Table 12: Human accuracies (%) by difficulty level, averaged over all scenes and annotators.

The monotonic drop from easy to hard indicates that the programmatic difficulty control in VideoCogQA (e.g., more objects, more complex trajectories, longer temporal dependencies) matches human-perceived difficulty. Humans still do not reach $100\%$, mainly due to occasional miscounting, high-speed or ambiguous trajectories, and visually similar grid configurations.

### A.5 DATASET DISTRIBUTION AND GAME-RELATED CAPABILITIES

Table 13 summarizes the distribution of videos and QA pairs across the six cognitive abilities and associated scenes in VideoCogQA. We further report model performance on three game-related capabilities (temporal reasoning, spatial reasoning, counting) in scenes S7–S9. Table 14 shows the

| Ability | Scene(s) | #Videos | #QA Pairs (easy/medium/hard) |
|---|---|---|---|
| Object Perception (OP) | S1 | 80 | 328 (109 / 109 / 110) |
| Action Perception (AP) | S2, S3 | 160 | 656 (218 / 218 / 220) |
| Temporal Reasoning (TR) | S4 | 80 | 328 (109 / 109 / 110) |
| Spatial Reasoning (SR) | S5, S6 | 160 | 656 (218 / 218 / 220) |
| GameWorld (GP) | S7, S8, S9 | 240 | 984 (327 / 327 / 330) |
| Full-modal (FP) | S10 | 80 | 328 (109 / 109 / 110) |
| Total | – | **800** | **3,280** |

Table 13: Distribution of videos and QA pairs in VideoCogQA across cognitive abilities and scenes. The benchmark is deliberately balanced across abilities and difficulty levels.

accuracies of representative VideoLLMs in the Game-related videos. The results reveal substantial headroom even for strong models.

| Model | Temporal | Spatial | Counting |
|---|---|---|---|
| VideoLLaVA | 34.4 | 33.7 | 25.4 |
| Qwen2-VL-7B | 34.4 | 47.6 | 33.3 |
| Qwen2-VL-2B | 25.0 | 31.3 | 28.6 |
| LLaVA-NeXT-Video-34B | 37.5 | 39.9 | 14.3 |
| Qwen2-VL-72B | 28.1 | 52.9 | 39.6 |
| Qwen2.5-VL-72B | 32.0 | 53.9 | 42.0 |
| Gemini | 56.0 | 47.6 | 25.7 |
| GPT-4o | 67.0 | 59.6 | 55.7 |

Table 14: Accuracies on three game-related capabilities (temporal reasoning, spatial reasoning, counting) for representative video-language models on the GameWorld scenes (S7–S9) in VideoCogQA.

## A.6 CORRELATION WITH REAL-WORLD BENCHMARKS

To assess both the diagnostic value and practical relevance of VideoCogQA, we analyze its relationship to natural-video benchmarks at two levels: (i) benchmark-level, between overall scores on VideoCogQA, VideoMME, LongVideoBench (LVB), and MLVU; and (ii) ability-level, between per-ability scores on VideoCogQA (OP, AP, TR, SR, GP) and the corresponding categories in VideoMME. We consider three Qwen2.5-VL models (3B, 7B, 72B) and, for each model, collect the above scores under a consistent evaluation setting of 2 FPS and at most 64 frames per video.

Table 17 reports benchmark-level Pearson correlations between VideoCogQA and the three natural-video benchmarks, using the overall scores from the three model scales. Table 15 lists the paired ability-wise accuracies on VideoCogQA and VideoMME (left/right within each cell), and Table 16 summarizes the resulting ability-level correlations. In both cases, the correlations are near-perfect: the cognitive dimensions measured by VideoCogQA closely track those in VideoMME, and higher performance on VideoCogQA reliably predicts stronger performance on real-world long-video benchmarks. Despite its synthetic, game-like nature, VideoCogQA thus aligns tightly with natural-video performance and serves as a practical diagnostic benchmark for VideoLLMs.

## A.7 TRANSFER TO NATURAL-VIDEO QA AFTER TUNING ON VIDEOCOGQA

To examine whether training on the synthetic tasks in VideoCogQA transfers to natural-video QA, we fine-tune Qwen2.5-VL-7B and Qwen2.5-VL-3B on the training split of VideoCogQA using LoRA (denoted as "Qwen2.5-VL-7B + VideoCogQA Tuning" and "Qwen2.5-VL-3B + VideoCogQA Tuning"). At the *benchmark level*, Table 18 shows that, after tuning, Qwen2.5-VL-7B

| Ability | Qwen2.5-VL-3B | Qwen2.5-VL-7B | Qwen2.5-VL-72B |
|---|---|---|---|
| OP | 43.1 / 55.0 | 47.8 / 62.5 | 48.8 / 67.0 |
| AP | 46.2 / 52.0 | 53.5 / 58.5 | 59.1 / 60.0 |
| TR | 51.2 / 48.0 | 57.0 / 56.0 | 56.8 / 57.0 |
| SR | 45.9 / 68.1 | 50.7 / 71.5 | 51.9 / 77.0 |
| GP | 43.9 / 30.6 | 44.3 / 36.2 | 45.2 / 34.7 |
| VideoCogQA (Avg) | 46.3 | 50.9 | 54.1 |
| VideoMME (Avg) | 54.7 | 60.5 | 64.2 |
| LVB (Avg) | 51.6 | 57.2 | 61.6 |
| MLVU (Avg) | 46.2 | 48.2 | 48.2 |

Table 15: Ability-based performance on VideoCogQA (left in each cell) and VideoMME (right) for three Qwen2.5-VL model scales. The bottom rows report average scores on VideoCogQA, VideoMME, LongVideoBench (LVB), and MLVU.

| Ability | OP | AP | TR | SR | GP |
|---|---|---|---|---|---|
| Correlation $r$ | 0.997 | 0.998 | 0.998 | 0.999 | 0.995 |

Table 16: Pearson correlation coefficients between ability scores on VideoCogQA and VideoMME across three Qwen2.5-VL model scales.

improves from 60.5 to 61.7 on VideoMME, from 57.2 to 58.5 on LongVideoBench, and from 48.2 to 50.6 on MLVU, yielding an average gain of $+1.6$ points. Qwen2.5-VL-3B improves from 54.7 to 59.8 on VideoMME and from 46.2 to 48.5 on MLVU, with an average gain of $+2.3$ points.

At the *ability level*, Tables 19 and 20 break down the VideoMME results by category and show that the largest improvements occur on temporal, spatial, and counting questions (e.g., temporal: $40.1 \rightarrow 47.8$ and counting: $30.6 \rightarrow 35.8$ for Qwen2.5-VL-3B; counting: $36.2 \rightarrow 45.2$ for Qwen2.5-VL-7B). Taken together, these benchmark-level and ability-level gains indicate that VideoCogQA provides measurable transfer to natural-video QA, especially for spatiotemporal and symbolic reasoning skills.

## A.8   FULL RESULTS

We report random-guessing accuracy for each scene in VideoCogQA in Table 21. The values are computed under the assumption that the model always outputs a valid option uniformly at random. In practice, some VideoLLMs fail to output a valid answer choice (e.g., they produce a free-form description instead of one of the listed options). We treat such outputs as incorrect. The reported accuracies are therefore often *lower* than the nominal random baseline, which assumes that a valid option is always produced. We present the complete performance results of the models across different scenes and difficulty levels in Table 32. These results provide a comprehensive comparison of model capabilities, highlighting performance variations under various cognitive tasks. This detailed analysis offers deeper insights into the strengths and weaknesses of each model when handling complex video understanding tasks.

## A.9   USAGE OF LLMS IN PAPER WRITING

In this paper, we have utilized Large Language Models (LLMs) to assist with various aspects of the writing process. Specifically, LLMs were used primarily for improving grammar, refining wording, and enhancing the clarity of certain sections of the text. They also assisted in drafting and polishing parts of the manuscript. We explicitly state that LLMs were used in these capacities during the preparation of the paper. We ensure that the content, ideas, and results presented in this paper are our own, and any contributions made by the LLMs were solely aimed at improving language quality and readability. The authors take full responsibility for the accuracy and integrity of the paper's content, ensuring that no misrepresentation or falsehoods were introduced through the use of LLMs.

| Pair | Correlation $r$ |
|------|-----------------|
| VideoCogQA Avg $\leftrightarrow$ VideoMME Avg | 0.995 |
| VideoCogQA Avg $\leftrightarrow$ LongVideoBench Avg | 0.997 |
| VideoCogQA Avg $\leftrightarrow$ MLVU Avg | 0.967 |

Table 17: Pearson correlation coefficients between average scores on VideoCogQA and three natural-video benchmarks (VideoMME, LongVideoBench, MLVU) using three Qwen2.5-VL model scales.

| Model / Benchmark | VideoMME | LongVideoBench | MLVU | Avg |
|-------------------|----------|----------------|------|-----|
| Qwen2.5-VL-7B | 60.5 | 57.2 | 48.2 | 55.3 |
| Qwen2.5-VL-7B + VideoCogQA Tuning | 61.7 | 58.5 | 50.6 | 56.9 |
| Qwen2.5-VL-3B | 54.7 | 51.6 | 46.2 | 50.8 |
| Qwen2.5-VL-3B + VideoCogQA Tuning | 59.8 | 51.0 | 48.5 | 53.1 |

Table 18: Overall accuracies on three natural-video QA benchmarks before and after fine-tuning Qwen2.5-VL-7B and Qwen2.5-VL-3B using VideoCogQA. The "+ VideoCogQA Tuning" indicates models fine-tuned on the synthetic benchmark.

The authors take full responsibility for the content of the manuscript, including any text generated or polished by the LLM. We have ensured that the LLM-generated text adheres to ethical guidelines and does not contribute to plagiarism or scientific misconduct.

| Category | Qwen2.5-VL-3B | Qwen2.5-VL-3B + VideoCogQA Tuning |
|---|---|---|
| Object | 57.7 | 63.0 |
| Action | 52.5 | 56.5 |
| Temporal | 40.1 | 47.8 |
| Spatial | 68.2 | 73.6 |
| Counting | 30.6 | 35.8 |
| Information | 71.5 | 75.9 |
| OCR | 64.8 | 67.6 |

Table 19: Category-level accuracies on VideoMME for Qwen2.5-VL-3B before and after fine-tuning using VideoCogQA. Fine-tuning on the synthetic benchmark yields consistent improvements, particularly in the temporal, spatial, and counting categories.

| Category | Qwen2.5-VL-7B | Qwen2.5-VL-7B + VideoCogQA Tuning |
|---|---|---|
| Object | 64.5 | 65.2 |
| Action | 59.0 | 58.2 |
| Temporal | 44.0 | 45.7 |
| Spatial | 71.8 | 72.7 |
| Counting | 36.2 | 45.2 |
| Information | 76.8 | 74.9 |
| OCR | 65.5 | 69.1 |

Table 20: Ability-level accuracies on VideoMME for Qwen2.5-VL-7B before and after fine-tuning on VideoCogQA. The largest improvement appears in counting, while temporal and spatial categories also benefit from the synthetic training.

| Scene | S1 | S2 | S3 | S4 | S5 | S6 | S7 | S8 | S9 | S10 |
|---|---|---|---|---|---|---|---|---|---|---|
| Random (%) | 33.3 | 33.3 | 33.3 | 30.3 | 32.7 | 28.9 | 28.9 | 30.3 | 33.3 | 33.3 |

Table 21: Random-guessing accuracy (%) per scene.

---

### Scene 1: Chameleon Grid

---

Which scene contains the largest number of {color} {shape} objects?
Which shape appears most frequently in the first column of Scene {i}?
How many {size} {color} {shape} objects appear across all scenes?
What is the total count of {size} {color} {shape} objects across all scenes?
In Scene {i}, which {attribute} is most common among the {size} objects?
In which scene does the grid contain the highest number of objects with the same {attribute}?
In Scene {i}, are there more {size_1} objects or {size_2} objects?
Is there a row in Scene {i} where all objects share the same {size} but vary in {color} or {shape}?
Between Scene {i} and Scene {j}, which shape appears more frequently: {shape_1} or {shape_2}?
Does Scene {i} contain any rows where all objects share the same {color} but differ in {shape}?

---

### Scene 2: Action Arena

---

How many objects are performing the {action_type} action?
How many distinct actions are performed by the {object}?
Which action has more objects performing it: {action_type1} or {action_type2}?
Which action is most frequently performed across all objects?
Which objects are currently performing the {action_type} action?
Which {action_type} actions appear more than {n} times?
What action is the {object} performing?
Are there any {shape} objects performing the {action_type} action?
Is there any {color} object performing the {action_type} action?

---

### Scene 3: Straight Paths

---

Which object moves the fastest: {object_a}, {object_b}, {object_c}, or {object_d}?
Which object moves more slowly: {object_a} or {object_b}?
How many times does the {object} change direction after hitting a wall?
How many objects initially move upward or downward when the movement begins?
How many objects initially move left or right when the movement begins?
Is the {object} moving left or right before it first hits a wall?
Is the {object} moving up or down before it first hits a wall?

---

### Scene 4: Time Sequence

---

What object appears immediately after the object {object}?
How long does the {object} appear?
At what timestamp does the first {object} appear?
What is the color of the object displayed after object {object}?
What is the total duration for which {object} is displayed after {time}?
At what time does the red circle first appear?
In what sequence do the objects appear after {time}, including {time} itself?
For how long does the blue circle appear after {time}, including {time} itself?
Did the blue circle appear at any time after {time}?
Did any object appear after {time}?
Are there any objects that appear both before and after {time}?

---

### Scene 5: Flash Grid

---

What is the shape and color of the first object that appears?

In which cell position does the first object appear?
Is there any {shape} located in row {i}?
Is there any {shape} located in column {j}?
Is there any {color} object located in row {i}?
Is there any {color} object located in column {j}?
In which row does {object} appear most frequently?
In which column does {object} appear most frequently?
How many unique cell positions are occupied by objects throughout the video?
For the {i}-th occurrence of the object, what is the object in the adjacent cell to the {direction}?

### Scene 6: 3D Navigator

Which edge of the cube is the red ball traverse most frequently by the ball?
Describe the path of movement taken by the ball.
On which edge does the small ball start moving?
On which edge does the ball conclude its movement?
How many times does the ball reach a vertex during its movement?
How many unique edges does the ball traverse throughout the video?

### Scene 7: Sky Battle

How many enemies are destroyed by the player?
At what timestamp does the player defeat the first enemy?
What is the movement pattern of the player's plane?
What is the maximum number of enemies visible on screen at any time?
How many enemies appear throughout the video?
How many times does the player's plane change direction?
Does the player's plane survive until the end of the scene?

### Scene 8: Maze Runner

Describe the player's full sequence of movements.
How many steps did the player take to solve the maze?
How many times did the player move upward or downward?
How many times did the player move left or right?
What is the length of the shortest path to solve the maze?
Is there a scene where the player fails to solve the maze within a specified number of steps?

### Scene 9: Tic-Tac-Toe Game

Which player made the first move in the game?
Which grid positions remained unplayed by the end of the game?
Which player won the game?
What was the final move made in the game?
What is the total number of moves made during the game?
How many moves did each player make?
Did either player win by completing a diagonal sequence?

### Scene 10: Note Matcher

Which musical note is associated with the appearance of the {object}?
How many different notes were played in total?
Which object appeared most frequently?

What was the first note played?
Was the note "{note}" played at any point?
How many times did the {note} occur alongside a {color} {shape}?
How many times does a {color} object appear alongside a {note}?

Table 31: QA templates for the ten video scenarios in VideoCogQA.

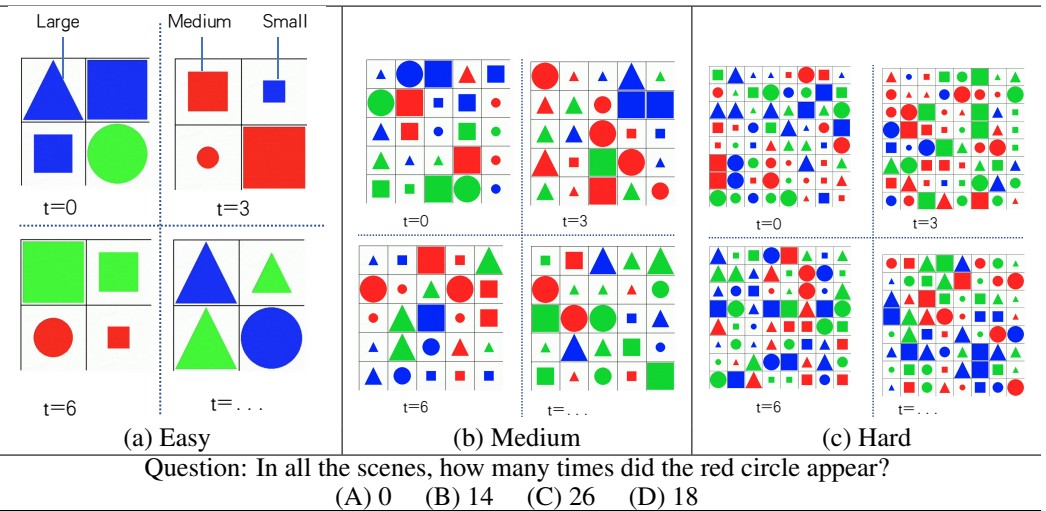

(a) Easy    (b) Medium    (c) Hard

Question: In all the scenes, how many times did the red circle appear?
(A) 0    (B) 14    (C) 26    (D) 18

Figure 10: Case of the 'Chameleon Grid' scene at different difficulty levels with the corresponding QA example.

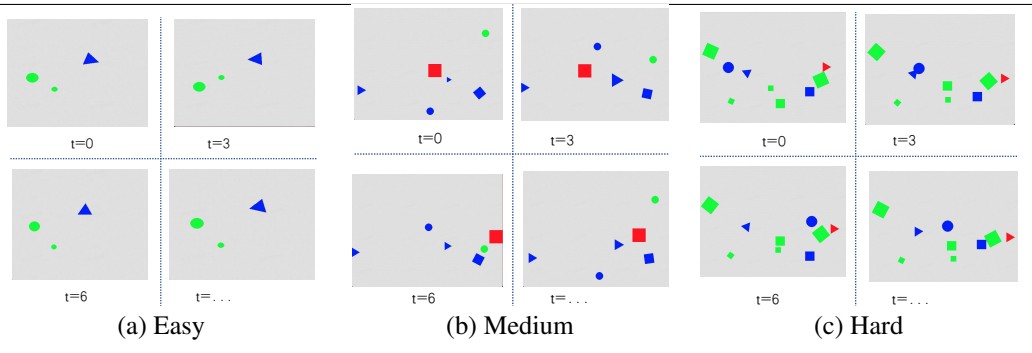

(a) Easy    (b) Medium    (c) Hard

Question: How many objects are performing the 'bounce' action in this scene?
(A) 0    (B) 2    (C) 6    (D) 8

Figure 11: Case of the 'Action Arena' scene at different difficulty levels with the corresponding QA example.

| Method | Difficulty | OP | AP | | TR | SR | | GP | | | FP |
|---|---|---|---|---|---|---|---|---|---|---|---|
| | | S1 | S2 | S3 | S4 | S5 | S6 | S7 | S8 | S9 | S10 |
| **MiniCPM-V** | Easy | 34.7 | 54.5 | 50.5 | 53.5 | 47.3 | 36.0 | 43.3 | 30.0 | 46.0 | 70.0 |
| | Medium | 26.0 | 48.5 | 36.5 | 50.5 | 30.0 | 40.0 | 23.3 | 26.7 | — | 43.3 |
| | Hard | 24.0 | 45.5 | 31.0 | 39.5 | 19.3 | 28.0 | 20.0 | 23.3 | — | 50.0 |
| **Video-LLaMA2** | Easy | 41.3 | 61.0 | 46.5 | 50.0 | 49.3 | 20.0 | 26.7 | 33.3 | 52.0 | 70.0 |
| | Medium | 29.3 | 52.0 | 26.5 | 53.0 | 31.3 | 18.0 | 23.3 | 20.0 | — | 56.7 |
| | Hard | 23.3 | 38.5 | 27.5 | 42.0 | 28.7 | 18.0 | 30.0 | 30.0 | — | 30.0 |
| **Video-LLaVA** | Easy | 42.7 | 21.0 | 36.0 | 27.5 | 40.0 | 28.0 | 23.3 | 33.3 | 38.0 | 66.7 |
| | Medium | 37.3 | 19.5 | 46.5 | 19.0 | 41.3 | 18.0 | 16.7 | 20.0 | — | 53.3 |
| | Hard | 41.3 | 22.5 | 40.0 | 23.0 | 31.3 | 18.0 | 10.0 | 23.3 | — | 60.0 |
| **LLaVA-NEXT-Video-34B** | Easy | 29.3 | 48.5 | 53.5 | 44.0 | 42.0 | 62.0 | 56.7 | 10.0 | 33.3 | 70.0 |
| | Medium | 26.0 | 40.0 | 42.5 | 45.0 | 18.7 | 54.0 | 36.7 | 6.7 | — | 56.7 |
| | Hard | 30.0 | 37.5 | 32.0 | 28.0 | 8.0 | 58.0 | 20.0 | 20.0 | — | 50.0 |
| **LLaVA-NEXT-Video-7B** | Easy | 31.3 | 25.5 | 38.0 | 22.5 | 38.0 | 16.0 | 6.7 | 6.7 | 15.3 | 56.7 |
| | Medium | 17.3 | 21.5 | 32.5 | 25.5 | 39.3 | 16.0 | 16.7 | 16.7 | — | 43.3 |
| | Hard | 12.7 | 20.5 | 21.5 | 15.0 | 24.0 | 24.0 | 13.3 | 13.3 | — | 40.0 |
| **InternLM-XComposer-2.5** | Easy | 53.3 | 47.0 | 52.0 | 46.5 | 34.0 | 20.0 | 13.3 | 40.0 | 35.3 | 70.0 |
| | Medium | 26.7 | 36.0 | 46.5 | 48.5 | 51.3 | 22.0 | 10.0 | 16.7 | — | 63.3 |
| | Hard | 28.0 | 31.5 | 38.0 | 38.5 | 44.0 | 18.0 | 3.3 | 20.0 | — | 50.0 |
| **Qwen2-VL-72B** | Easy | 61.3 | 65.0 | 67.0 | 64.5 | 76.0 | 52.0 | 40.0 | 63.3 | 62.0 | 83.3 |
| | Medium | 48.7 | 59.5 | 62.5 | 56.0 | 56.0 | 34.0 | 33.3 | 23.3 | — | 80.0 |
| | Hard | 45.3 | 50.0 | 50.5 | 50.0 | 50.0 | 40.0 | 23.3 | 26.7 | — | 66.7 |
| **Gemini-2.0-Flash** | Easy | 52.5 | 52.0 | 52.5 | 71.5 | 53.3 | 38.0 | 30.0 | 56.7 | 55.5 | 76.7 |
| | Medium | 37.6 | 41.5 | 41.5 | 70.0 | 51.3 | 38.0 | 26.7 | 40.0 | — | 73.3 |
| | Hard | 32.7 | 42.0 | 31.5 | 64.0 | 45.3 | 36.0 | 26.7 | 20.0 | — | 63.3 |
| **GPT-4o** | Easy | 48.0 | 72.0 | 55.0 | 67.0 | 71.3 | 74.0 | 70.0 | 60.0 | 72.0 | 76.6 |
| | Medium | 45.3 | 66.5 | 54.5 | 64.0 | 70.7 | 66.0 | 56.7 | 53.0 | — | 73.3 |
| | Hard | 39.3 | 56.0 | 45.0 | 53.0 | 62.6 | 66.0 | 53.3 | 43.0 | — | 60.0 |

Table 32: Performance of more models across scenes and difficulty levels.

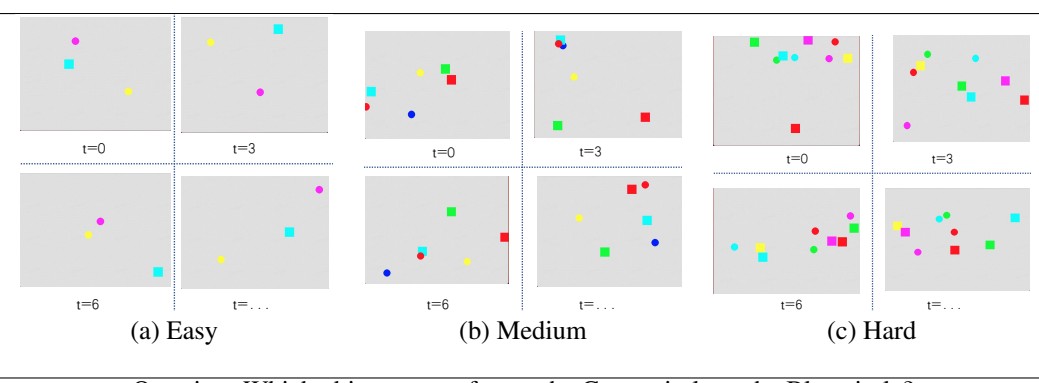

Question: Which object moves faster: the Green circle or the Blue circle?
(A) Green circle     (B) Blue circle     (C) Both are equal

Figure 12: Case of the 'Straight Paths' scene at different difficulty levels with the corresponding QA example.

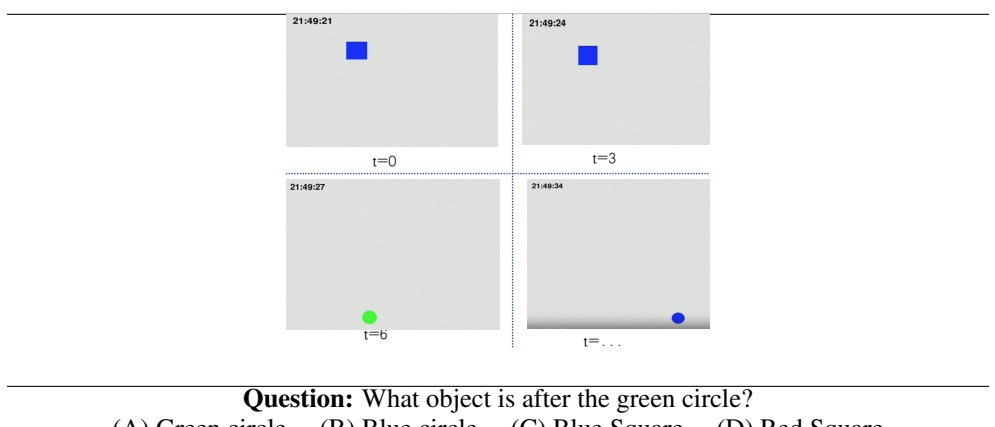

**Question:** What object is after the green circle?
(A) Green circle     (B) Blue circle     (C) Blue Square     (D) Red Square

Figure 13: Example of the 'Time Sequence' scene at the Easy difficulty level with the corresponding QA example.

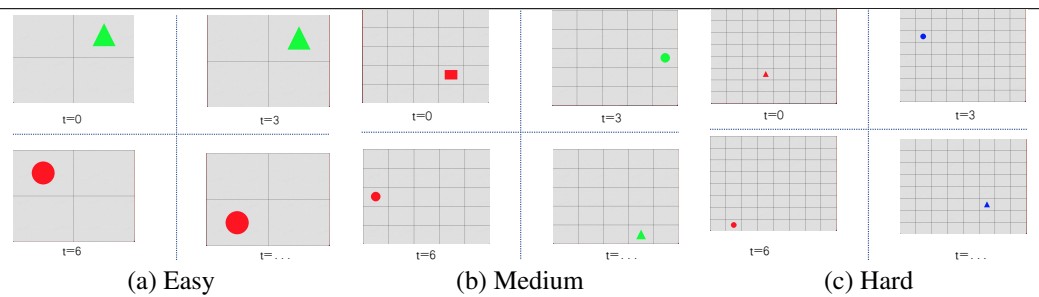

For the second occurrence of the object, what is the object in the adjacent cell to the left?
(A) Green circle     (B) Red circle     (C) Both are equal

Figure 14: Case of the ' Flash Grid' scene at different difficulty levels with the corresponding QA example.

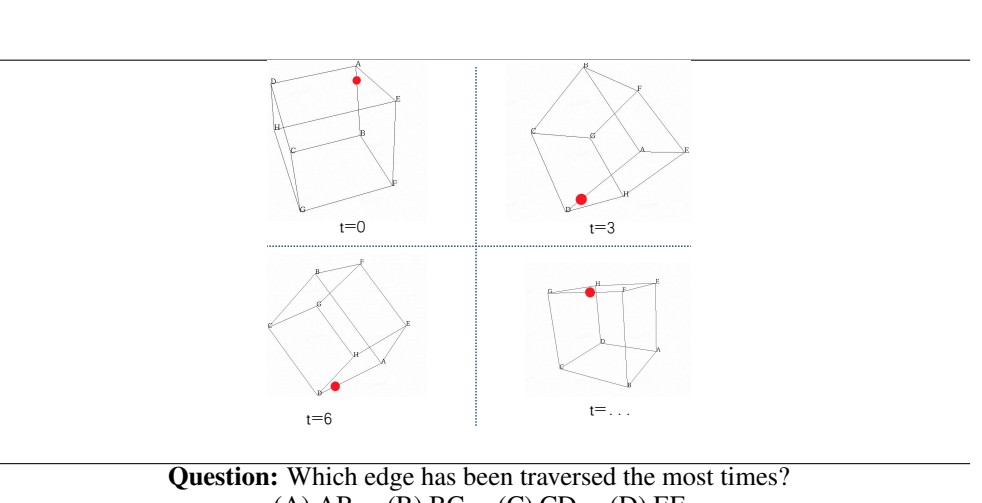

**Question:** Which edge has been traversed the most times?
(A) AB     (B) BC     (C) CD     (D) EF

Figure 15: Example of the '3D Navigator' scene at the Easy difficulty level with the corresponding QA example.

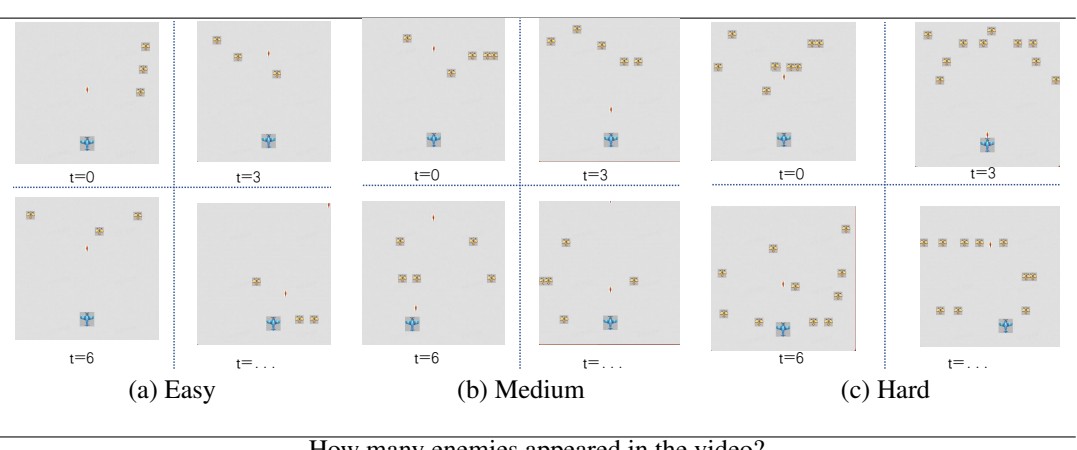

(a) Easy                    (b) Medium                    (c) Hard

How many enemies appeared in the video?
(A) 6     (B) 8     (C) 12     (D) 3

Figure 16: Case of the 'Sky Battle' scene at different difficulty levels with the corresponding QA example.

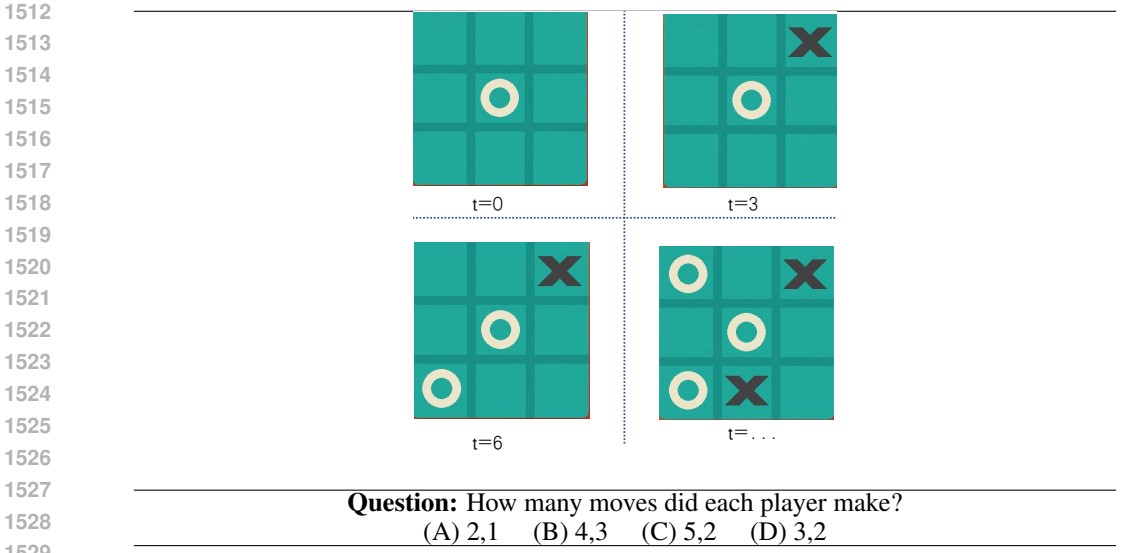

**Question:** How many moves did each player make?
(A) 2,1    (B) 4,3    (C) 5,2    (D) 3,2

Figure 17: Example of the 'Tic-Tac-Toe Game' scene at the Easy difficulty level with the corresponding QA example.

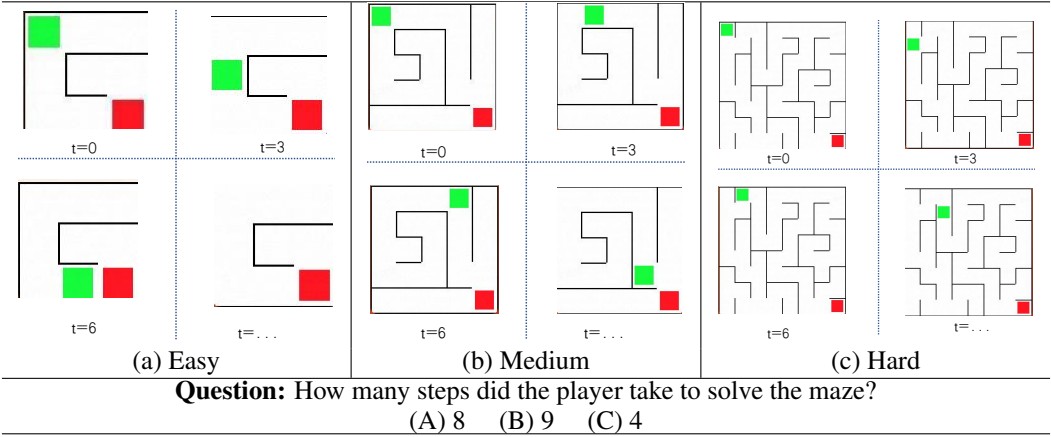

(a) Easy                    (b) Medium                    (c) Hard

**Question:** How many steps did the player take to solve the maze?
(A) 8    (B) 9    (C) 4

Figure 18: Example of the ' Maze Runner' scene at different difficulty levels, along with the corresponding QA example.

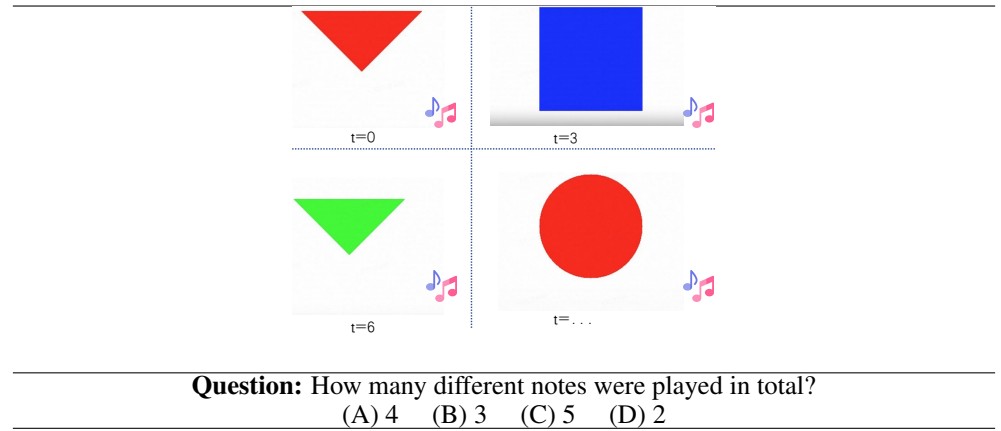

**Question:** How many different notes were played in total?
(A) 4    (B) 3    (C) 5    (D) 2

Figure 19: Example of the 'Note Matcher' scene at the Easy difficulty level with the corresponding QA example.

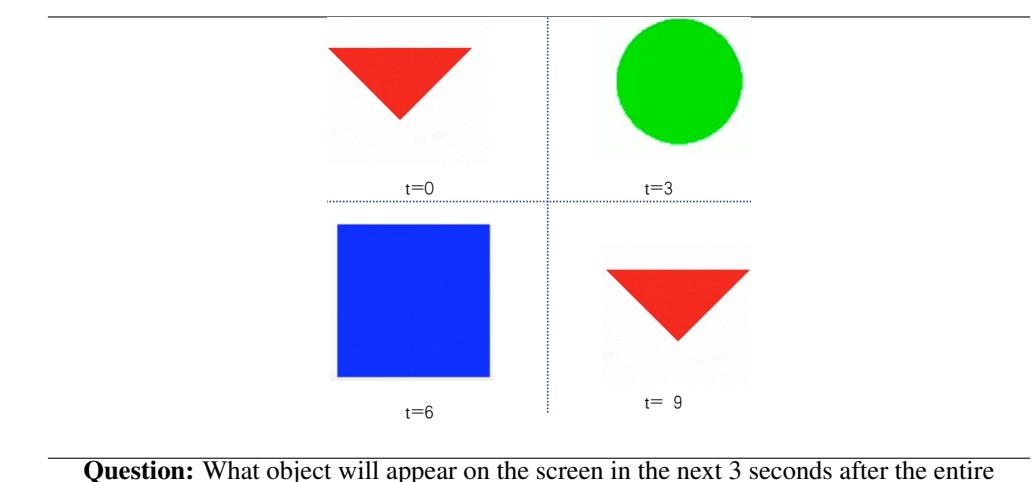

**Question:** What object will appear on the screen in the next 3 seconds after the entire video?
(A) Green circle     (B) Blue circle     (C) Green square     (D) Blue square

Figure 20: Test videos and Q&As can be automatically generated, such as predicting the next object based on previous patterns, evaluating its pattern recognition ability.

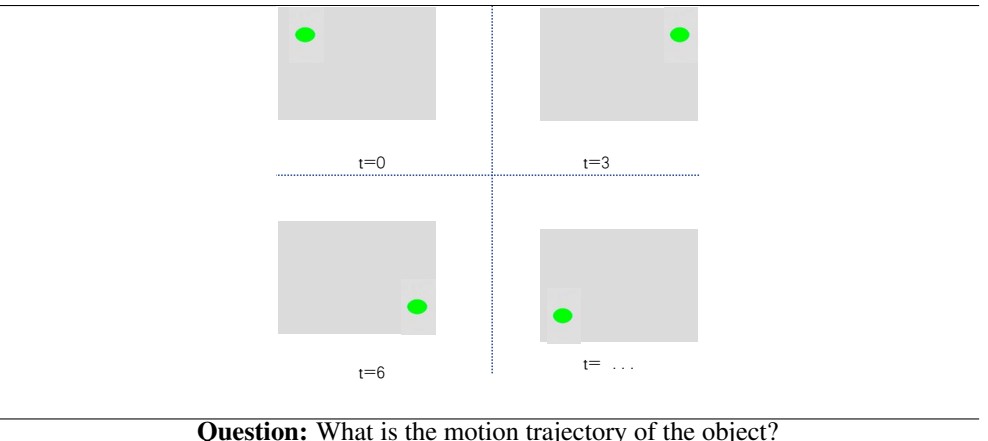

**Question:** What is the motion trajectory of the object?
(A) Circle     (B) Square     (C) Triangle     (D) Line

Figure 21: Test videos and Q&As can be automatically generated, such as analyzing the motion trajectory of an object based on videos, evaluating its object tracking ability.

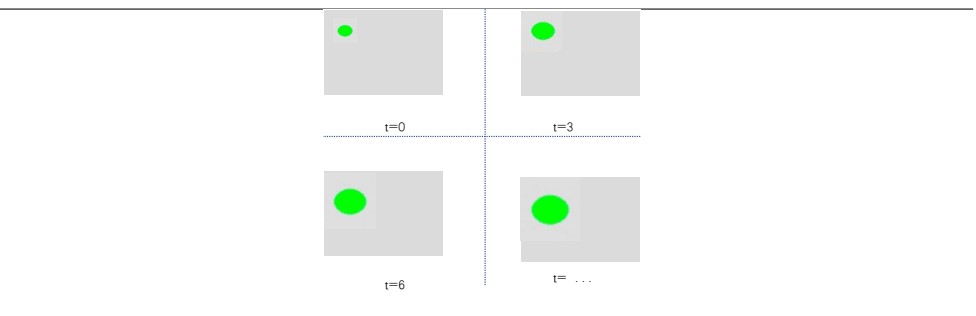

**Question:** What changes occur in the video content?
(A) The green ball gets bigger     (B) The green ball gets smaller     (C) No change     (D) The green ball gets bigger and then smaller

Figure 22: Test videos and Q&As can be automatically generated, such as evaluating the visual perception of object changes, assessing its fine-grained visual perception ability.

