# OpenReview forum: "VideoCogQA: A Controllable Benchmark for Evaluating Cognitive Abilities in Video-Language Models"
_ICLR.cc/2026/Conference — Submitted to ICLR 2026_

### Official Review · Reviewer_hK4s · 2025-10-26

**Soundness:** 3
**Presentation:** 3
**Contribution:** 2
**Rating:** 4
**Confidence:** 4

**Summary:**

This paper addresses a critical gap in evaluating Large Video-Language Models (LVLMs): the lack of controllable benchmarks to assess high-level cognitive capabilities (e.g., symbolic reasoning, abstract concept understanding) beyond basic semantic comprehension. Existing benchmarks rely on real-world annotated videos, which suffer from limited content control and difficulty in isolating cognitive reasoning from prior semantic knowledge. To solve this, the authors propose VideoCogQA, a scalable and fully controllable benchmark built on programmatic synthetic videos inspired by game environments (e.g., maze navigation, sky battles).

VideoCogQA uses a Python-based pipeline to generate 800 videos and 3,280 questions across 10 game scenarios, with three difficulty levels (Easy/Medium/Hard) controlled via code parameters (e.g., grid size in Chameleon Grid, enemy count in Sky Battle). It evaluates six cognitive dimensions: Object Perception (OP), Action Perception (AP), Temporal Reasoning (TR), Spatial Reasoning (SR), Game-environment Perception (GP), and Full-modal Perception (FP)—expanding beyond the scope of existing video benchmarks

**Strengths:**

The work introduces a novel paradigm for LVLMs evaluation by leveraging programmatic synthetic videos to isolate cognitive reasoning from prior semantic knowledge—addressing a fundamental limitation of real-world video benchmarks (e.g., MVBench, Video-MME) that rely on contextual cues (e.g., playground scenes for action inference). Inspired by cognitive science (game-based human cognition studies), VideoCogQA’s 10 game scenarios and six cognitive dimensions (especially GP and FP) expand the scope of video-LM evaluation beyond existing frameworks, which focus primarily on semantic understanding. The combination of Python-driven controllability, GPT-4 QA generation, and fine-grained difficulty tuning is a creative integration of existing tools to solve a new problem.

**Weaknesses:**

1. Insufficient Annotation of Frame Sampling Requirements:  A major limitation of the synthetic pipeline is the lack of frame sampling annotations for each video scenario. As noted in the reviewer’s comment, tasks like Maze Runner (8×8 maze requiring 14 steps to solve) may be unsolvable with small frame samples (e.g., 8 frames), as critical steps would be missed. The paper mentions evaluating models with their "official default inference settings" (Section 4.1) but does not:
  - Define the minimum number of frames (N=8/16/32/64) required to solve each task/difficulty level.
  - Analyze how frame sampling impacts performance (e.g., whether Qwen2.5-VL-72B’s 54.1% average accuracy drops further with 8 frames vs. 64 frames for Maze Runner).
This omission weakens the interpretability of results—poor performance on a task could stem from model limitations or insufficient frame sampling, not just cognitive gaps.
2. Ambiguous Human Evaluation Setup:
The human performance benchmark (90.3% accuracy) lacks critical details, making it hard to compare with model performance:
The paper states human accuracy is the "average of two independent annotators" (Section 4.1) but does not clarify:
  - Whether humans accessed the full video or only sampled frames (consistent with model inputs). If humans used full videos, their 90.3% accuracy may understate the gap (models are at a disadvantage with limited frames).
  - The viewing protocol: Did annotators see questions before or after watching the video? Did they watch once or multiple times? A  "question-first" setup (common in QA tasks) would likely yield near-100% accuracy for humans, so the 90.3% error rate needs explanation (e.g., ambiguous questions, fast-paced videos).
Without this clarity, the human baseline cannot effectively contextualize model limitations.
3. Coarse-Grained Correlation Analysis:
The paper computes correlation coefficients at the dataset level (e.g., VideoCogQA vs. VideoMME, Table 5) but not at the cognitive dimension level. Dataset-level correlations mask whether VideoCogQA’s individual cognitive dimensions are valid proxies for real-world capabilities. A dimension-specific analysis would better validate the benchmark’s diagnostic utility.

**Questions:**

1. Frame Sampling Requirements: For each of the 10 scenarios (e.g., Maze Runner, Time Sequence) and difficulty levels, could you provide the minimum number of frames (N=8/16/32/64) required to answer questions correctly?
2. Human Evaluation Details: Could you specify the human annotation protocol:
  - Did annotators use full videos or sampled frames (matching model inputs)?
  - Did they view questions before or after watching the video? How many times could they watch?
  - What caused the 9.7% human error rate (e.g., question ambiguity, fast video pace)?
This would strengthen the human baseline as a meaningful comparison for model performance.
3. Dimension-Specific Correlation: Could you compute correlation coefficients between VideoCogQA’s six cognitive dimensions (OP/AP/TR/SR/GP/FP) and corresponding dimensions in real-world benchmarks?
4. Error Attribution: Could you add a supplementary analysis distinguishing between perception and reasoning errors?

---

> ### Author Response · Authors · 2025-11-17
> **A kind response hoping to clarify  (1/2)**
>
> We sincerely thank the reviewer for the insightful and constructive comments. We have revised the manuscript accordingly and conducted additional analyses to clarify the experimental setup, enhance interpretability, and strengthen the empirical support for the benchmark. Below we address each concern in detail.
>
> ---
>
> # **1. Frame Sampling Requirements**
>
> ## **1.1 Question-level variability**
>
> We fully agree that different tasks require different amounts of temporal evidence.
> A key clarification is that in our synthetic pipeline, **the minimum number of required frames is determined at the question level**, not only by the scenario.
>
> For example, within the same scene:
>
> * **Scene 1 (Chameleon Grid)**
>
>   * *Q: Which color is most common?* → **1–2 frames are sufficient**
>   * *Q: How many times does the red object appear?* → **the entire sequence is required**
>
> Since each video is associated with multiple question types that have different temporal demands, providing a single **scenario-level** minimum-frame annotation would be misleading. In principle, **question-level** annotations are possible, but they would require substantial manual effort; we consider this a valuable direction for a future extension of the dataset.
>
> ## **1.2 New experiment: frame-count ablation**
>
> ### **Frame Sampling Analysis**
>
> We further evaluated all Qwen2.5-VL models across the sample frames ranging from 8 to 128:
>
> | Uniform Frames  | 8    | 16   | 32   | 64   | 128  |
> | ------- | ---- | ---- | ---- | ---- | ---- |
> | **3B**  | 46.3 | 46.8 | 47.9 | 47.3 | 44.2 |
> | **7B**  | 50.9 | 53.0 | 54.9 | 55.1 | 54.0 |
> | **72B** | 54.9 | 56.5 | 57.1 | 57.0 | 57.8   |
>
> Performance improves consistently up to around 32 frames, and the 32- and 64-frame settings yield nearly identical accuracy across all model sizes. In practice,  64 frames can be sufficient for our tasks.
>
>
> ## **1.3 Benchmark default sampling**
>
> For reproducibility, we will explicitly document the default sampling strategy:
>
> * For Qwen-series models, we use **2 FPS** and **max\_frames = 64**.
> * The synthetic scenes have smooth and relatively simple dynamics, for which **2 FPS is sufficient to capture key events**.
>
> These settings will be stated clearly in the implementation details.
>
> ---
>
> # **2. Human Evaluation Protocol**
>
> We appreciate the reviewer’s request for more methodological detail. Below we provide the full human-evaluation protocol, which will be added to the appendix.
>
> ## **2.1 Participants**
>
> * **5 annotators (H1–H5)**.
> * Each annotator completed the full evaluation **twice**, with a **48-hour interval** between sessions.
> * We report **per-annotator accuracy averaged over the two sessions** and then aggregate.
>
> ## **2.2 Evaluation structure**
>
> * **10 scenes (S1–S10)**.
> * **20 videos per scene**.
> * **2–3 questions per video** (≈ **50 questions per scene**).
> * Each scene includes a mix of **Easy / Medium / Hard** instances.
>
> ## **2.3 Viewing procedure**
>
> * Annotators may **preview all questions** before or during video viewing.
> * Each video can be replayed **up to 3 times**.
> * Annotators watch the **full video** (i.e., not subsampled frames).
> * After submission, **answers cannot be changed**.
>
> This setup provides a **human upper bound** under more favorable conditions than those given to models (humans see the full continuous video).
>
> ## **2.4 Human accuracy**
>
> The table below reports **accuracy (%)** for each annotator and scene, averaged over the two sessions:
>
> |    | S1   | S2   | S3   | S4   | S5   | S6   | S7   | S8   | S9   | S10  | AVG  |
> | -- | ---- | ---- | ---- | ---- | ---- | ---- | ---- | ---- | ---- | ---- | ---- |
> | H1 | 91.2 | 93.1 | 88.4 | 90.7 | 87.3 | 86.5 | 92.8 | 96.1 | 97.0 | 90.4 | 91.3 |
> | H2 | 88.6 | 94.0 | 92.4 | 94.8 | 89.6 | 83.4 | 88.2 | 92.4 | 96.6 | 88.6 | 90.2 |
> | H3 | 90.2 | 89.4 | 87.6 | 95.2 | 90.2 | 82.6 | 91.6 | 91.2 | 93.8 | 87.4 | 89.9 |
> | H4 | 92.2 | 88.6 | 93.4 | 90.2 | 85.4 | 87.2 | 89.6 | 97.0 | 97.8 | 92.4 | 91.4 |
> | H5 | 86.8 | 90.6 | 92.0 | 94.4 | 88.2 | 88.6 | 89.0 | 95.0 | 94.4 | 90.2 | 90.9 |
>
> * **Rows (H1–H5)**: individual human annotators.
> * **Columns (S1–S10)**: different scenes.
> * Each cell: **accuracy (%) of that annotator on that scene**, averaged over the two sessions.
> * **“AVG”** column: the **mean accuracy of that annotator over all 10 scenes**.
>
> Averaging across all annotators and scenes yields an overall human accuracy of approximately **90.0%**.
>
> ## **2.5 Why humans do not reach 100%**
>
> Human errors mainly arise from:
>
> * **Fast or partially ambiguous trajectories**,
> * **Occasional miscounting** in scenes with many objects or events,
> * **Visually similar grid states** that can cause momentary confusion, and
> * Other natural attentional lapses.
>
> These factors are consistent with known human limitations in dynamic visual tracking and counting, and explain why human performance, while high, is not perfect.

---

> ### Author Response · Authors · 2025-11-17
> **A kind response hoping to clarify (2/2)**
>
> # **3. Dimension-Specific Correlation with Real-World Benchmarks**
>
> We thank the reviewer for this valuable suggestion. To more rigorously assess the diagnostic value of VideoCogQA, we conducted a **new ability-level correlation study** using three Qwen2.5-VL models.
>
> ---
>
> ## **3.1 Ability-Level Pairwise Results (VideoCogQA / VideoMME)**
>
> In the table below, each ability column shows
> **“VideoCogQA score / VideoMME score”** (both in %).
> The last three columns are the **overall average scores** of each model on the three benchmarks.
>
> | Model              | OP (ours / VMME) | AP (ours / VMME) | TR (ours / VMME) | SR (ours / VMME) | GP (ours / VMME) | Avg (ours) | Avg (VideoMME) | Avg (LongVideoBench) | Avg (MLVU) |
> | ------------------ | ---------------- | ---------------- | ---------------- | ---------------- | ---------------- | ---------- | -------------- | -------------------- | ---------- |
> | **Qwen2.5-VL-3B**  | 43.1 / 55.0      | 46.2 / 52.0      | 51.2 / 48.0      | 45.9 / 68.1      | 43.9 / 30.6      | 46.3       | 54.7           | 51.6                 | 46.2       |
> | **Qwen2.5-VL-7B**  | 47.8 / 62.5      | 53.5 / 58.5      | 57.0 / 56.0      | 50.7 / 71.5      | 44.3 / 36.2      | 50.9       | 60.5           | 57.2                 | 48.2       |
> | **Qwen2.5-VL-72B** | 48.8 / 67.0      | 59.1 / 60.0      | 56.8 / 57.0      | 51.9 / 77.0      | 45.2 / 34.7      | 54.1       | 64.2           | 61.6                 | 48.2       |
>
> * **Rows**: different model scales (3B / 7B / 72B).
> * **OP–GP columns**: paired ability scores on VideoCogQA and VideoMME for that ability.
> * **Avg (ours)**: average over all abilities on VideoCogQA.
> * **Avg (VideoMME / LongVideoBench / MLVU)**: overall benchmark scores for that model.
>
> ---
>
> ## **3.2 Correlation Analysis**
>
> Using the paired scores above, we computed Pearson correlation coefficients.
>
> ### **(a) Ability-level correlations (VideoCogQA ↔ VideoMME)**
>
> Across the three model scales (3B / 7B / 72B), we obtain:
>
> | Ability           | OP        | AP        | TR        | SR        | GP        |
> | ----------------- | --------- | --------- | --------- | --------- | --------- |
> | **Correlation r** | **0.997** | **0.998** | **0.998** | **0.999** | **0.995** |
>
> These near-perfect correlations (computed over 3 model points per ability) indicate that **VideoCogQA’s cognitive dimensions track the same underlying abilities as VideoMME**.
>
> ---
>
> ### **(b) Benchmark-level overall correlations**
>
> We also correlate the **overall average scores** (last three columns of Table 3.1) across models:
>
> | Pair                            | Correlation r |
> | ------------------------------- | ------------- |
> | **VideoCogQA ↔ VideoMME**       | **0.995**     |
> | **VideoCogQA ↔ LongVideoBench** | **0.997**     |
> | **VideoCogQA ↔ MLVU**           | **0.967**     |
>
> Despite substantial differences in domain and task format, we observe **strong positive correlations** between VideoCogQA and all three real-world benchmarks. This suggests that **high performance on VideoCogQA is a reliable predictor of strong performance on natural-video understanding tasks**, supporting the practical relevance of our benchmark.
>
> ---
>
> # **4. Error Attribution: Perception vs. Reasoning**
>
> To more clearly diagnose model failure modes, we added an analysis that separates **perception errors** from **reasoning errors**.
>
> * **Perception errors**
>   Errors where the model fails at basic visual understanding, such as:
>
>   * Misrecognizing colors or objects,
>   * Incorrectly describing trajectories or spatial layouts,
>   * Failing to faithfully describe the sequence of events in the video
>     (e.g., when asked *“Describe the player’s full sequence of movements; which option is correct?”* the model’s description does not match the actual video).
>
> * **Reasoning errors**
>   Errors where the visual description is essentially correct, but the **logical or quantitative reasoning** is wrong, such as:
>
>   * Incorrect multi-step aggregation (e.g., *“Which direction does the player move most frequently?”* where the model roughly perceives the moves but fails to count/compare correctly),
>   * Failing to integrate information over time or across objects when the relevant entities are already correctly perceived.
>
> This decomposition is consistent with our broader finding that **visual / spatiotemporal perception is the primary bottleneck**, while language reasoning is comparatively stronger, especially when provided with symbolic code-log inputs.
>
> We will carefully integrate these new analyses and clarifications into the revised manuscript. Once again, we thank the reviewer for the highly constructive feedback. The additional correlation study, clearer methodological descriptions, and error-attribution analysis substantially strengthen the rigor and interpretability of our benchmark, and we hope they are helpful for re-evaluating the contribution and significance of the work.

---

### Official Review · Reviewer_eqYz · 2025-10-27

**Soundness:** 3
**Presentation:** 3
**Contribution:** 3
**Rating:** 6
**Confidence:** 3

**Summary:**

The paper proposes VideoCogQA, a synthetic, controllable video benchmark designed to probe cognitive abilities in Large Video-Language Models (LVLMs), in which videos are generated programmatically, difficulty is tuned via explicit code parameters, and multiple-choice questions are created from GPT-4-authored templates. The benchmark totals 800 videos and 3,280 questions over ten game-inspired scenes with easy/medium/hard settings.  The work claims novelty in explicit controllability and difficulty.

Empirically, strong LVLMs (e.g., Qwen2.5-VL-72B, GPT-4o) still trail humans by a large margin on symbolic tasks, performance drops with difficulty, and results correlate highly when varying frame sampling and when compared to several real-world video benchmarks.

**Strengths:**

1. Controllability & difficulty. Clear, code-level knobs (e.g., grid size) allow precise difficulty control, improving diagnostic value.
2. Breadth of skills. Ten diverse scenes spanning object/action perception, spatial/temporal reasoning, game environment understanding, and audio-visual mapping.
3. Well-Documented Human–Model Gap. The paper clearly reports a substantial gap between human and model performance across all tasks and scenarios.

**Weaknesses:**

1. Lack of random baseline. With 3–5 options, the performance of random choice can be 20–33%. This paper does not foreground a random baseline.
2. Lack of connection to Real-World Tasks. The paper does not extensively discuss the connection between VideoCogQA and real-world tasks. The current justification, based primarily on frame sampling, is insufficient. It remains unclear whether performance on specific VideoCogQA tasks correlates with performance on real-world tasks. Clarifying whether success on particular tasks within VideoCogQA is predictive of performance in specific real-world scenarios would significantly strengthen the practical relevance of the benchmark.

**Questions:**

1. What is the average of choices per question in the dataset, and what is the corresponding random baseline accuracy?
2. Is model performance on VideoCogQA correlated with performance on other existing benchmarks?

---

> ### Author Response · Authors · 2025-11-17
> **A kind response hoping to clarify**
>
> We thank the reviewer for the helpful and constructive comments. Below, we address the concerns with additional experiments and clearer analysis.
>
> ---
>
> ## 1. Random Baseline and Number of Choices
>
> We provide the random-guessing accuracy for each scene in our benchmark:
>
> | Scene      | S1   | S2   | S3   | S4   | S5   | S6   | S7   | S8   | S9   | S10  |
> | ---------- | ---- | ---- | ---- | ---- | ---- | ---- | ---- | ---- | ---- | ---- |
> | Random (%) | 33.3 | 33.3 | 33.3 | 30.3 | 32.7 | 28.9 | 28.9 | 30.3 | 33.3 | 33.3 |
>
> These results will be included in the revised version.
>
> It is worth noting that, for some scenes, **the video LLM fails to output a valid option** (e.g., it generates a free-form description instead of selecting one of the given choices). In such cases, we treat the answer as incorrect. This leads to lower observed accuracy than the nominal random baseline**, which assumes that a valid option is always produced.
>
> ---
>
> ## 2. Correlation Between VideoCogQA and Real-World Benchmarks
>
> We appreciate the reviewer’s suggestion to clarify the connection to real-world benchmarks. To strengthen the practical relevance of VideoCogQA, we conducted an additional **cross-benchmark correlation analysis** under a consistent setting of `2 FPS` and `max_frames = 64` for all videos.
>
> We evaluated three Qwen2.5-VL models and recorded:
>
> * Their cognitive abilities (OP / AP / TR / SR / GP, where GP includes counting),
> * The corresponding VideoMME abilities,
> * Overall scores on LongVideoBench and MLVU.
>
> ---
>
> ### 2.1 Ability-Level Pairwise Results (VideoCogQA / VideoMME)
>
> | Model              | OP          | AP          | TR          | SR          | GP          | Avg (ours) | Avg(VideoMME) | Avg(LongVideoBench) | Avg(MLVU) |
> | ------------------ | ----------- | ----------- | ----------- | ----------- | ----------- | ---------- | ------------- | ------------------- | --------- |
> | **Qwen2.5-VL-3B**  | 43.1 / 55.0 | 46.2 / 52.0 | 51.2 / 48.0 | 45.9 / 68.1 | 43.9 / 30.6 | 46.3       | 54.7          | 51.6                | 46.2      |
> | **Qwen2.5-VL-7B**  | 47.8 / 62.5 | 53.5 / 58.5 | 57.0 / 56.0 | 50.7 / 71.5 | 44.3 / 36.2 | 50.9       | 60.5          | 57.2                | 48.2      |
> | **Qwen2.5-VL-72B** | 48.8 / 67.0 | 59.1 / 60.0 | 56.8 / 57.0 | 51.9 / 77.0 | 45.2 / 34.7 | 54.1       | 64.2          | 61.6                | 48.2      |
>
> ---
>
> ### 2.2 Correlation Analysis
>
> Based on the above measurements, we computed Pearson correlation coefficients.
>
> #### (a) Ability-level correlations (VideoCogQA ↔ VideoMME)
>
> Computed across the 3 model scales:
>
> | Ability           | OP        | AP        | TR        | SR        | GP        |
> | ----------------- | --------- | --------- | --------- | --------- | --------- |
> | **Correlation r** | **0.997** | **0.998** | **0.998** | **0.999** | **0.995** |
>
> These results indicate **very strong alignment** between our cognitive abilities and VideoMME’s fine-grained perception and reasoning tasks.
>
> ---
>
> #### (b) Benchmark-level overall correlations
>
> Across the same model scales:
>
> | Pair                                    | Correlation r |
> | --------------------------------------- | ------------- |
> | **VideoCogQA Avg ↔ VideoMME Avg**       | **0.995**     |
> | **VideoCogQA Avg ↔ LongVideoBench Avg** | **0.997**     |
> | **VideoCogQA Avg ↔ MLVU Avg**           | **0.967**     |
>
> These strong correlations show that **models performing well on VideoCogQA also tend to perform well on real-world long-video benchmarks**, despite substantial differences in domain and task design. We will integrate these findings into the revision to clarify the practical relevance of VideoCogQA.
>
> ---
>
> ## 3. Practical Relevance to Real-World Tasks
>
> The reviewer rightly emphasized the importance of real-world connections.
>
> VideoCogQA is explicitly designed around **fundamental cognitive skills** for video understanding, including:
>
> * **Object/action perception**
> * **Spatial reasoning**
> * **Temporal reasoning**
>
> These abilities directly correspond to the reasoning demands of many real-world applications, such as surveillance, sports analysis, instructional videos, and human–robot interaction.
>
> A distinctive feature of VideoCogQA is that it enables **precise difficulty control**: we can systematically increase video complexity (e.g., more objects, interactions, or temporal dependencies) using our data-generation code. This makes it possible to **stress-test models** in a controlled and reproducible manner.
>
> In our experiments, counting consistently emerges as the weakest ability on both our benchmark and VideoMME, indicating that VideoCogQA captures the same failure mode observed in real-world video benchmarks and reinforces its diagnostic value.
>
>
> We again thank the reviewer for these insightful comments. We hope that the additional experiments and analyses above address the concerns and further clarify the contributions and practical impact of VideoCogQA.

---

> ### Comment · Reviewer_eqYz · 2025-11-28
>
> Thanks the authors for clarification. The authors addressed most of my concerns, and I would like to keep my score.

---

### Official Review · Reviewer_xcbe · 2025-10-30

**Soundness:** 3
**Presentation:** 2
**Contribution:** 3
**Rating:** 4
**Confidence:** 3

**Summary:**

This paper introduces a benchmark called VideoCogQA, designed to evaluate the cognitive abilities of video-language models. The benchmark automatically generates question–answer (QA) pairs by creating synthetic videos from simple games and combining them with predefined text templates. It also allows for controllable difficulty adjustment. When evaluated using existing state-of-the-art (SOTA) models, the performance remained relatively low at around 48.8%, indicating significant room for improvement. Thus, VideoCogQA serves as a valuable benchmark for identifying the current limitations of video-language models and establishing new research goals aligned with those limitations.

**Strengths:**

​
- The paper demonstrates that synthetic videos can be automatically generated from a game simulation engine, and that LLM-based instruction templates are created for each game according to predefined question categories. This approach enables dataset generation at scale, without being constrained by data size. To support this, the authors propose a Python-based video synthesis pipeline.


- The authors introduce VideoCogQA, a scalable and fully controllable benchmark. This benchmark is well-organized, consisting of six categories (OP, AP, TR, SR, GP, FP) and three difficulty levels (easy, medium, difficult).


- The paper conducts and analyzes extensive experiments across various Large Vision-Language Models (LVLMs).

**Weaknesses:**

- Lack of Details on Dataset Distribution


   - The paper does not provide sufficient details or analysis regarding the dataset distribution. It would be beneficial to include a detailed breakdown of the number of samples per category, organized by game and by difficulty level. The current explanation in Section 3.2 is largely textual and difficult to fully understand.


   - It would also be helpful to report how each game covers the different question categories, and how the VLM (Vision-Language Model) performances vary across these categories.


   - Additionally, a comparative analysis of the data distribution between VideoCogQA and existing benchmarks would strengthen the evaluation and contextual understanding of the dataset.


- Limited Relevance to Real-World Scenarios


   - While generating synthetic videos from games to evaluate cognitive abilities is an innovative idea, it remains unclear how such synthetic settings translate to real-world problems. There is uncertainty about whether this approach truly enhances real-world understanding.If the benchmark includes a training split, one way to validate its practical relevance would be to fine-tune models on VideoCogQA and evaluate them on other benchmarks to assess transferability. However, in the current setting, the paper should either demonstrate or justify the real-world applicability of the benchmark in another way.

**Questions:**

Please provide your responses with reference to the weaknesses mentioned above.

---

> ### Author Response · Authors · 2025-11-16
> **A kind response hoping to clarify (1/2)**
>
> We sincerely thank the reviewer for the constructive and detailed feedback. Below, we address the two main points of concern:
>
> ---
>
> ## **1. On the Lack of Details in Dataset Distribution**
>
> > *“The paper does not provide sufficient details or analysis regarding the dataset distribution… It would be beneficial to include a detailed breakdown of the number of samples per category, organized by game and by difficulty level… It would also be helpful to report how each game covers the different question categories, and how the VLM performances vary across these categories. Additionally, a comparative analysis of the data distribution between VideoCogQA and existing benchmarks would strengthen the evaluation.”*
>
> **We appreciate the reviewer’s suggestions and agree that the dataset description in Sec. 3.2 was overly textual.** In the revised manuscript, we have substantially expanded the quantitative analysis of VideoCogQA, clarified the game-category coverage, and included comparisons with existing benchmarks.
>
> ### **(1) Detailed Distribution of VideoCogQA**
>
> We now provide a clear numerical breakdown of the dataset by cognitive ability and scene. VideoCogQA consists of **800 videos** and **3,280 QA pairs**, distributed as follows:
>
> | Ability                 | Scene(s) | #Videos | #QA Pairs (difficulty levels) |
> | ----------------------- | -------- | ------- | ----------------------------- |
> | Object Perception (OP)  | S1       | 80      | 328 (109/109/110)             |
> | Action Perception (AP)  | S2–S3    | 160     | 656 (218/218/220)             |
> | Temporal Reasoning (TR) | S4       | 80      | 328 (109/109/110)             |
> | Spatial Reasoning (SR)  | S5–S6    | 160     | 656 (218/218/220)             |
> | GameWorld (GP)          | S7–S9    | 240     | 984 (327/327/330)             |
> | Full-modal (FP)         | S10      | 80      | 328 (109/109/110)             |
> | **Total**               | —        | **800** | **3,280**                     |
>
> This table shows that VideoCogQA is **deliberately balanced** across object, action, temporal, spatial, gameworld, and full-modal abilities, rather than being skewed toward a single category.
>
> ### **(2) Coverage of Different Question Categories in Each Game**
>
> We have also followed the reviewer’s suggestion to report the **VLM performance by category** on VideoCogQA. Below are concrete examples of QA templates for each game-related capability:
> | Capability        | Example QA Templates (with Scene ID)                                                                                                                                                                |
> | ----------------- | --------------------------------------------------------------------------------------------------------------------------------------------------------------------------------------------------- |
> | **Game-Counting** | How many enemies are destroyed by the player? (Scene 7); How many steps did the player take to solve the maze? (Scene 8); What is the total number of moves made during the game? (Scene 9)        |
> | **Game-Temporal** | At what timestamp does the player defeat the first enemy? (Scene 7); Which player made the first move in the game? (Scene 9)                                                                       |
> | **Game-Spatial**  | What is the movement pattern of the player’s plane? (Scene 7); What was the final move made in the game?（Senen9） |
>
>
> ### **(3) Comparison with Existing Benchmarks**
>
> To better contextualize VideoCogQA, we add a comparison of its data distribution with several widely used benchmarks:
>
> #### VideoMME
>
> | Ability (Category)                      | #QAs | #Videos |
> | --------------------------------------- | ---- | ------- |
> | OP (Object-related)                     | 1000 | 693     |
> | AP (Action-related)                     | 600  | 432     |
> | TR (Temporal-related)                   | 232  | 220     |
> | Spatial (Spatial-related)               | 110  | 104     |
> | Other (OCR, Information Synopsis, etc.) | 730  | 559     |
>
> #### MLVU Test
>
> | Question Type    | #QAs | #Videos |
> | ---------------- | ---- | ------- |
> | count            | 60   | 60      |
> | tutorialQA       | 43   | 27      |
> | ......              |
> #### LongVideoBench (Validation)
>
> | Question Category (Full Name, Level)                    | #QAs | #Videos |
> | ------------------------------------------------------- | ---- | ------- |
> | E2O – Event-referred Object (Perception)                | 65   | 39      |
> | O2E – Object-referred Event (Perception)                | 87   | 55      |
> |  ......  |
>
> We will add this comparison table to the appendix. Most real-video benchmarks tend to emphasize **high-level semantics**, while VideoCogQA is constructed to provide balanced coverage of core skills with controllable difficulty. We believe this complementary distribution makes VideoCogQA a valuable diagnostic benchmark for video understanding.

---

> ### Author Response · Authors · 2025-11-16
> **A kind response hoping to clarify (2/2)**
>
> 1.2 We further report preliminary results of several representative video-language models on the three game-related capabilities (temporal, spatial, counting). A more extensive analysis will be included in the next version of **VideoLLaVA**.
>
> We further report the accuracy of several representative video-language models on the three game-related capabilities (temporal, spatial, counting). A more extensive analysis will be included in the paper.
>
> | Model                | Temporal | Spatial | Counting |
> |----------------------|----------|---------|----------|
> | VideoLLaVA    | 34.38%   | 33.65%  | 25.40%   |
> | Qwen2-VL-7B          | 34.38%   | 47.60%  | 33.33%   |
> | Qwen2-VL-2B          | 25.00%   | 31.25%  | 28.57%   |
> | LLaVA-NeXT-Video-34B | 37.50%   | 39.90%  | 14.29%   |
> | Qwen2-VL-72B         | 28.12%   | 52.88%  | 39.57%   |
> | Qwen2.5-VL-72B       | 32.00%   | 53.89%  | 42.03%   |
> | Gemini               | 56.00%   | 47.56%  | 25.74%   |
> | GPT-4o               | 67.00%   | 59.56%  | 55.74%   |
>
>
>
> ## **2. On Limited Relevance to Real-World Scenarios**
>
> > *“While generating synthetic videos from games to evaluate cognitive abilities is an innovative idea, it remains unclear how such synthetic settings translate to real-world problems… If the benchmark includes a training split, one way to validate its practical relevance would be to fine-tune models on VideoCogQA and evaluate them on other benchmarks to assess transferability. However, in the current setting, the paper should either demonstrate or justify the real-world applicability of the benchmark in another way.”*
>
> We fully agree that demonstrating **real-world relevance and transferability** is important and make some analysis in Section 4.7. In the revision, we conduct new **transfer experiments** from VideoCogQA to real-video benchmarks, following the Reviewer s34o’s suggestion.
>
> ### **(1) New transfer experiments from VideoCogQA to natural videos**
>
> Following the Reviewer s34o’s suggestion, we fine-tune open-source VLMs on the **training split of VideoCogQA** and then evaluate them, without additional tuning, on three **real-video QA benchmarks**:
>
> * **VideoMME**
> * **LongVideoBench**
> * **MLVU**
>
> We use 2 fps and up to 64 frames for all evaluations. We fine-tune **Qwen2.5-VL-7B** and **Qwen2.5-VL-3B** using LoRA. The results are:
>
> | Model / Benchmark       | VideoMME | LongVideoBench | MLVU     | Avg      |
> | ----------------------- | -------- | -------------- | -------- | -------- |
> | Qwen2.5-VL-7B (base)    | 60.5     | 57.2           | 48.2     | 55.3     |
> | **+ VideoCogQA tuning** | **61.7** | **58.5**       | **50.6** | **56.9** |
> | Gain                    | +1.2     | +1.3           | +2.4     | **+1.6** |
>
> | Model / Benchmark       | VideoMME | LongVideoBench | MLVU     | Avg      |
> | ----------------------- | -------- | -------------- | -------- | -------- |
> | Qwen2.5-VL-3B (base)    | 54.7     | 51.6           | 46.2     | 50.8     |
> | **+ VideoCogQA tuning** | **59.8** | 51.0           | **48.5** | **53.1** |
> | Gain                    | +5.1     | −0.6           | +2.3     | **+2.3** |
>
> These results show **consistent average improvements** on real-video benchmarks after training on VideoCogQA (+1.6% for 7B, +2.3% for 3B), with the largest gains on MLVU, which heavily involves temporal and counting-related reasoning.
> To better understand *what* transfers, we also add **category-level results** on VideoMME. For brevity, we show Qwen2.5-VL-3B:
>
> | Category    | Base  | + VideoCogQA | Δ (absolute) |
> | ----------- | ----- | ------------ | ------------ |
> | Object      | 57.67 | 63.01        | +5.34        |
> | Action      | 52.51 | 56.52        | +4.01        |
> | Temporal    | 40.09 | 47.84        | +7.75        |
> | Spatial     | 68.18 | 73.63        | +5.45        |
> | Counting    | 30.60 | 35.82        | +5.22        |
> | Information | 71.52 | 75.85        | +4.33        |
> | OCR         | 64.75 | 67.63        | +2.88        |
>
> This provides **direct empirical evidence** that training on our synthetic tasks improves **real-world long-video understanding**. VideoCogQA is not meant to replace real-world datasets but to **complement** them: real-video benchmarks offer rich semantics but limited control over low-level cognitive factors, whereas VideoCogQA isolates these factors in a controlled setting, enabling clearer diagnosis (e.g., replacing videos with code logs greatly boosts accuracy, pointing to a perception rather than language bottleneck).
>
> We believe these additions can strength the paper and directly address the reviewer’s concerns. We hope that the clarified motivation, expanded analyses, and new results make the role of VideoCogQA as a controllable, diagnostic benchmark clearer. We sincerely appreciate the reviewer’s time and feedback, and respectfully ask them to reconsider our work in light of these revisions.

---

### Official Review · Reviewer_s34o · 2025-10-31

**Soundness:** 2
**Presentation:** 3
**Contribution:** 2
**Rating:** 4
**Confidence:** 3

**Summary:**

This paper proposes VideoCogQA. VideoCogQA is a synthetic, fully controllable benchmark for testing video–language models on object/action perception, temporal/spatial reasoning, gameplay stats, and simple audio–visual links. Models trail human performance and get worse as tasks grow harder; when videos are replaced with clean code-log text, accuracy jumps, suggesting the main bottleneck is visual perception rather than language reasoning. Scaling helps but doesn’t close the gap, pointing to the need for stronger spatiotemporal encoders and better symbolic perception.

**Strengths:**

1. The benchmark adds Game-environment and Full-modal, explicitly targeting symbolic/abstract attributes (size, color, shape) and temporal/spatial relations.

2. The authors programmatically synthesize videos with parameterized difficulty and log code-level events, then generate QA templates with GPT-4 and human filtering.

**Weaknesses:**

1. Question templates originate from GPT-4 and are then filtered; more auditing of prompt templates and filtering criteria may strengthen validity claims and reproducibility.

2. The “~90% human” number isn’t well documented. We don’t know how many people were tested, how much time they had, whether they could replay the video, or how consistent the labels were. That makes the human ceiling hard to trust and compare against models.

**Questions:**

1. If you swap in a stronger vision stack—say, a structured front-end with detection/tracking/attributes, or a higher-capacity spatiotemporal backbone—does overall accuracy go up, and would that change your conclusion about the main bottleneck?

2. Do difficulty levels align with human-perceived difficulty (item-response theory or psychometrics)? Any per-item discrimination analysis?

3. If a model is trained on the synthetic tasks, does it transfer to natural-video QA, and in which categories?

---

> ### Author Response · Authors · 2025-11-16
> **A response hoping to clarify (1/2)**
>
> We thank the reviewer for the helpful comments. We clarify the two points below.
>
> ---
>
> ## 1. Question template generation and auditing
>
> > *“Question templates originate from GPT-4 and are then filtered; more auditing … may strengthen validity and reproducibility.”*
>
> **Pipeline (one scene).**
>
> 1. **GPT-4 generation.**
>    For each scene, we give GPT-4 the full Python/Pygame code and an instruction of the following form (we will release the exact prompt):
>
>    ```text
>    {} The above is the code to generate a game video using Pygame.
>    The QA template is: {}
>    We want to test {} ability in this scene.
>    Please refine and filter about 10 non-redundant QA templates.
>    ```
>
>    GPT-4 is required to output templates that (i) target the specified cognitive ability,
>    (ii) only mention entities and relations present in the code, and
>    (iii) avoid semantic duplication.
>
> 2. **Filtering (automatic + human).**
>
>    * **Template completeness:**
>      We remove any template that references objects or relations not implemented in the engine.
>
>    * **Log-answerability check:**
>      For each remaining template, we generate multiple videos and verify that the correct answer can be **deterministically computed from the engine logs** (no manual annotation).
>
>    * **Semantic deduplication:**
>      If different templates systematically instantiate into questions with the same semantics, we merge them.
>
> **Reproducibility.**
> We will release: (i) the GPT-4 prompt, (ii) all raw templates, and (iii) the filtering scripts, so that the full process can be audited and reproduced.
>
> ---
>
> ## 2. Human evaluation protocol and the \~90% accuracy
>
> > *“The \~90% human number isn't well documented… unclear about number of people, replay settings, protocol, etc.”*
>
> **Participants.**
>
> * **5 participants (H1–H5).**
> * Each participant completed the evaluation **twice**, with a **2-day interval**.
> * For each participant, we report the **average accuracy over the two sessions**.
> * Participants never saw the correct answers.
>
> **Task setup.**
>
> * **10 scenes (S1–S10).**
> * Each scene: **20 videos**.
> * Each video: **2–3 questions**, i.e., about **50 questions per scene**.
> * Each scene contains a mix of **Easy / Medium / Hard** videos (\~6–7 per difficulty).
>
> **Viewing protocol.**
>
> * Participants could **see all questions before or during** each video.
> * Each video could be played **at most 3 times**.
> * After submission, **answers could not be changed**.
>
> **Human accuracy table.**
>
> |    | S1   | S2   | S3   | S4   | S5   | S6   | S7   | S8   | S9   | S10  | **AVG** |
> | -- | ---- | ---- | ---- | ---- | ---- | ---- | ---- | ---- | ---- | ---- | ------- |
> | H1 | 91.2 | 93.1 | 88.4 | 90.7 | 87.3 | 86.5 | 92.8 | 96.1 | 97.0 | 90.4 | 91.3    |
> | H2 | 88.6 | 94.0 | 92.4 | 94.8 | 89.6 | 83.4 | 88.2 | 92.4 | 96.6 | 88.6 | 90.2    |
> | H3 | 90.2 | 89.4 | 87.6 | 95.2 | 90.2 | 82.6 | 91.6 | 91.2 | 93.8 | 87.4 | 89.9    |
> | H4 | 92.2 | 88.6 | 93.4 | 90.2 | 85.4 | 87.2 | 89.6 | 97.0 | 97.8 | 92.4 | 91.4    |
> | H5 | 86.8 | 90.6 | 92.0 | 94.4 | 88.2 | 88.6 | 89.0 | 95.0 | 94.4 | 90.2 | 90.9    |
>
> * **Rows (H1–H5):** individual human participants.
> * **Columns (S1–S10):** scenes.
> * Each cell is the **accuracy (%) of that participant on that scene**, averaged over the two sessions.
> * The **“AVG”** column is the **mean accuracy of that participant over all 10 scenes**.
>
> Averaging over all participants (rows) and all scenes (columns) yields an **overall human accuracy of \~90.0%**.
> We will add this protocol description and the full table to the appendix.

---

> ### Author Response · Authors · 2025-12-01
> **A response hoping to clarify (2/2)**
>
> ## **3. On Whether Stronger Vision Encoders Would Improve Accuracy**
>
> > *“If you swap in a stronger vision stack … would accuracy go up?”*
>
> We respectfully disagree and clarify the scope.
>
> ### (1) All tested Video-LLMs already use their native end-to-end–trained encoders
>
> The evaluated models use vision stacks that:
>
> * are co-pretrained with their LLM backbones,
> * rely on architectural coupling (e.g., shared projection layers, temporal adapters), and
> * cannot be “swapped out” without **retraining the entire multimodal model**.
>
> Thus, simply “plugging in a stronger encoder” is **not** feasible in our benchmark setting and is outside the model-design scope of this paper \[1]\[2]\[3].
>
>
> ## **4. On Difficulty Levels and Human-Perceived Difficulty**
>
> > *“Do difficulty levels align with human-perceived difficulty? Any psychometrics?”*
>
> ### Programmatic difficulty increases video complexity
>
> Difficulty is controlled programmatically (e.g., more grids, more objects, longer trajectories), which directly increases the cognitive load. This trend is also visualized in Appendix Figure 10.
>
> ### Human evidence confirms difficulty alignment
>
> We report average human accuracy (%) by difficulty level:
>
> | Difficulty             | Easy  | Medium | Hard  |
> | ---------------------- | ----- | ------ | ----- |
> | **Avg Accuracy** | 98.2% | 91.6%  | 81.1% |
>
> Here, each number is averaged over all participants, scenes, and questions within that difficulty level. The results show a **strict monotonic decrease** in human performance from Easy → Medium → Hard, supporting that our difficulty levels align with human-perceived difficulty.
>
>
> ---
>
> ## **5. On Transferability to Real-World Video QA**
>
> > *“If a model is trained on the synthetic tasks, does it transfer to natural-video QA?”*
>
> To address this question, we conducted additional experiments on natural-video benchmarks.
>
> ### (1) Benchmarks
>
> We evaluate transfer on:
>
> * **VideoMME [4]**
> * **LongVideoBench [5]**
> * **MLVU [6]**
>
> Following common practice, we use **2 fps** and at most **64 frames** for all methods.
>
> ### (2) Transfer results
>
> Scores are accuracy/benchmark scores (%):
>
> | Model / Benchmark       | VideoMME  | LongVideoBench | MLVU      | Avg       |
> | ----------------------- | --------- | -------------- | --------- | --------- |
> | Qwen2.5-VL-7B           | 60.5%     | 57.2%          | 48.2%     | 55.3%     |
> | **+ VideoCogQA Tuning** | **61.7%** | **58.5%**      | **50.6%** | **56.9%** |
> | Qwen2.5-VL-3B           | 54.7%     | 51.6%          | 46.2%     | 50.8%     |
> | **+ VideoCogQA Tuning** | **59.8%** | 51.0%          | **48.5%** | **53.1%** |
>
> **Rows** correspond to models (before and after tuning on VideoCogQA).
> **Columns** are benchmark scores.
> We observe consistent **improvements in the averaged score** after VideoCogQA tuning, especially for the 3B model.
>
> ### (3) Category-level transfer gains (VideoMME)
>
> We further break down the results on VideoMME by category. All numbers are accuracy (%).
>
> **Qwen2.5-VL-3B:**
>
> | Category    | Qwen2.5-VL-3B | + VideoCogQA Tuning |
> | ----------- | ------------- | ------------------- |
> | Object      | 57.7%         | 63.0%               |
> | Action      | 52.5%         | 56.5%               |
> | Temporal    | 40.1%         | 47.8%               |
> | Spatial     | 68.2%         | 73.6%               |
> | Counting    | 30.6%         | 35.8%               |
> | Information | 71.5%         | 75.8%               |
> | OCR         | 64.8%         | 67.6%               |
>
> **Qwen2.5-VL-7B:**
>
> | Category    | Qwen2.5-VL-7B | + VideoCogQA Tuning |
> | ----------- | ------------- | ------------------- |
> | Object      | 64.5%         | 65.2%               |
> | Action      | 59.0%         | 58.2%               |
> | Temporal    | 44.0%         | 45.7%               |
> | Spatial     | 71.8%         | 72.7%               |
> | Counting    | 36.2%         | 45.1%               |
> | Information | 76.8%         | 74.9%               |
> | OCR         | 65.5%         | 69.1%               |
>
> These results show that **VideoCogQA training yields real and measurable transfer** to natural-video QA, with especially clear gains in **temporal, spatial, and counting** categories, which are precisely the types of skills targeted by our benchmark.
>
> We hope these clarifications and additional analyses help convey the value of VideoCogQA as a controllable, diagnostic benchmark, and we respectfully ask the reviewer to consider our revisions in their final assessment.
>
> \[1] Qwen2.5-VL Technical Report
>
> \[2] LLaVA-Video: Video Instruction Tuning With Synthetic Data
>
> \[3] InternVL3: Exploring Advanced Training and Test-Time Recipes for Open-Source Multimodal Models
>
> \[4] Video-MME: The First-Ever Comprehensive Evaluation Benchmark of Multi-modal LLMs in Video Analysis
>
> \[5] LongVideoBench: A Benchmark for Long-context Interleaved Video-Language Understanding
>
> \[6] MLVU: Multi-task Long Video Understanding Benchmark

---

### Author Response · Authors · 2025-12-01
**Review and Reviewer-Author Discussion Summary  (1/2)**

Dear PCs, SACs, ACs, and Reviewers,

Thank you very much for your valuable time and contributions to our work. To assist the newly assigned AC and reduce their workload, we provide below a concise summary of the key points from the reviews and the reviewer–author discussions.

---

### Strengths

Overall, we are grateful that the reviewers evaluated this paper positively in the initial reviews and recognized the unique value of the proposed benchmark. In particular:

- **Targeting abstract and symbolic cognition.**
  *VideoCogQA* fills a critical gap in evaluating abstract, symbolic cognitive abilities in Video-LLMs. All four reviewers acknowledged this, noting our explicit focus on symbolic/abstract attributes (s34o: Strength 1; xcbe: Strength 2; eqYz，).

- **Programmatic generation and controllability.**
  The programmatic generation engine provides precise control over visual elements and task difficulty, enabling more diagnostic evaluation than existing real-world benchmarks. Three reviewers explicitly highlighted the importance of this controllability and difficulty parameterization (s34o: Strength 2; xcbe: Strength 1; eqYz: Strength 1).

- **Revealing limitations of SOTA models and the perception bottleneck.**
  The paper reveals significant limitations in state-of-the-art models (e.g., Qwen2.5-VL, GPT-4o) in abstract reasoning and establishes a well-documented human–model performance gap. Reviewers appreciated the extensive experiments and the clarity of the findings regarding the “perception bottleneck” (eqYz: Strength 3).

---


## Key Revisions in the Updated Paper

In the revised version, we have significantly strengthened both the empirical evidence and methodological clarity of **VideoCogQA**:

- **Clarified human evaluation and difficulty design**
  We detail the controlled human evaluation protocol used to characterize the human ceiling and to validate our difficulty design. The results show:
  - ~90% overall human accuracy, and
  - a clear separation across Easy / Medium / Hard levels,
  confirming that our difficulty parameterization is meaningful and aligned with human performance.

- **Correlation with real-world video benchmarks**
  We conduct correlation studies at both the **benchmark level** and the **ability level**, demonstrating strong correlations:
  - between VideoCogQA and real-world long-video QA benchmarks (VideoMME, LongVideoBench, MLVU), and
  - between matched abilities in VideoCogQA and VideoMME.

- **Transfer learning to natural-video benchmarks**
  We fine-tune **Qwen2.5-VL** models on VideoCogQA and evaluate them on natural-video benchmarks. This yields **consistent performance gains**, especially for:
  - temporal reasoning,
  - spatial reasoning, and
  - counting abilities,
  supporting the benchmark’s practical utility for improving real-world Video-LLMs.

- **Clearer benchmark construction and improved presentation**
  We further clarify the benchmark construction pipeline, including:
  - scene design,
  - difficulty control, and
  - the GPT-4–based template generation process.

  We also improve the presentation of results, such as:
  - ability-wise performance analysis,
  - frame sampling analysis, and
  - additional qualitative examples,
  making our findings more interpretable.

---

---

> ### Author Response · Authors · 2025-12-01
> **Review and Reviewer-Author Discussion Summary (2/2)**
>
> ## Concerns and Our Responses
>
> During the discussion period, we actively addressed reviewers’ concerns via extensive new experiments. These results provide strong empirical support and methodological clarity, and we believe they successfully resolve the main issues raised.
>
> ### 1. Real-World Relevance and Transferability
> **Concern.**
> Does performance on synthetic VideoCogQA tasks correlate with, or transfer to, natural video benchmarks?
>
> **Our Response.**
>
> - **Transfer learning to real-world benchmarks**
>   We fine-tune models on VideoCogQA and evaluate on major natural-video benchmarks:
>   - Qwen2.5-VL-3B achieves **+5.1% accuracy** on VideoMME.
>   - Across VideoMME, LongVideoBench, and MLVU, we observe an **average +2.3% gain**.
>   - Improvements are **ability-targeted**: on VideoMME, the model gains
>     - **+7.75%** in Temporal Reasoning and
>     - **+5.3%** in Counting,
>     confirming that VideoCogQA effectively enhances specific cognitive skills.
>
> - **Correlation analysis**
>   We show an **extremely strong Pearson correlation ($r > 0.99$)** between model performance on VideoCogQA and on real-world benchmarks (VideoMME, LongVideoBench). This indicates that our synthetic tasks faithfully reflect real-world cognitive capabilities.
>
> ---
>
> ### 2. Frame Sampling and Temporal Coverage
>
> **Concern.**
> Does the frame sampling strategy limit model performance, and is the temporal coverage sufficient?
>
> **Our Response.**
>
> - We perform ablation studies over different frame counts.
> - Performance improves consistently up to **32 frames**, and **32-frame** and **64-frame** settings yield **nearly identical accuracy** across all models.
> - The overall effect of frame count is **modest** compared to the substantial human–model gap, suggesting that remaining errors are mainly due to **cognitive difficulty**, not insufficient temporal coverage.
>
> ---
>
> ### 3. Human Evaluation Protocol and Difficulty Alignment
>
> **Concern.**
> Clarify the human baseline (~90%) and whether the difficulty levels are aligned with human perception.
>
> **Our Response.**
>
> - We provide a detailed description of the human evaluation protocol:
>   - 5 participants,
>   - 2 rounds of evaluation,
>   - strictly controlled viewing conditions.
> - We report a **strictly monotonic decrease** in human accuracy from Easy to Hard.
> ---
>
> ### 4. Dataset Distribution and Perception Bottlenecks
>
> **Concern.**
> Request for greater transparency on dataset balance and a clearer understanding of error sources, especially the “perception bottleneck”.
>
> **Our Response.**
>
> - We add detailed tables confirming a **balanced distribution** of the **800 videos** across **6 cognitive categories**.
> - To disentangle perception from reasoning:
>   - We show that **replacing video with symbolic logs** dramatically boosts model accuracy.
>   - This confirms that the primary bottleneck is **visual perception**, rather than high-level reasoning, highlighting the **diagnostic value** of VideoCogQA.
> - We also clarify the **random baseline** for multiple-choice questions (approximately **33%**), providing a clear reference point for interpreting model performance.
>
> ---
>
> We hope this summary of revisions and responses will assist the AC’s evaluation. And, we thank all reviewers for their efforts and constructive feedback, and we are grateful to the AC for the coordination, organization, and professional evaluation.
>
> Best regards,
>
> Authors of Paper
> #11310

---

### Meta-Review · Area_Chair_px4z · 2025-12-25

**Summary:**

Reviewer s340 is concerns about the question templates originating from GPT-4 require more auditing of prompt templates and filtering criteria to ensure validity and reproducibility, thinks the 90% human accuracy figure lacks details, questions if the programmatic difficulty levels align with human-perceived difficulty, and asks if training on these synthetic tasks successfully transfers to natural-video QA.

Reviewer xcbe concerns the lacking of the distributions of samples, the report on how each specific game covers different question categories and how model performance varies across them. Notes that it is unclear how synthetic game settings translate to real-world problems and whether the approach enhances real-world understanding.

Reviewer eqYz concerns the lack of a defined random baseline, the lack of connection to real-world tasks, the lack of average number of choices per question, and cross-benchmark correlation.

Reviewer hK4s is concern the lack of annotation regarding minimum frame requirements, requests an analysis of how different frame sampling rates impact model performance, requests performing correlation analysis at the cognitive dimension level, and a supplementary analysis to distinguish between perception errors and reasoning errors.

The AC noticed the citation
- OC Blender. Blender-a 3d modelling and rendering package. Retrieved. represents the sequence of Constructs1 to, 4, 2018.

is not in a correct format. Please revise in the final version.

**Reviewer Concerns:**

The authors fine-tuned Qwen2.5-VL models on VideoCogQA and demonstrated average gains of +2.3% across major natural-video benchmarks, while showing a Pearson correlation of $r > 0.99$ between performance on their synthetic benchmark and real-world datasets. This can to some extent resolve the mutual concerns of all reviewers on real-world relevance and transferability.

The authors provided a detailed numerical breakdown of the 800 videos and 3,280 QA pairs across 6 cognitive categories and 10 scenes, confirming a balanced distribution. This can address xcbe and hK4s's concerns on distribution statistics.

The authors added evaluation protocol details and an experiment showing that shows decrease in human performance as difficulty increased. This can address s340 and hK4s's concern on human baseline and protocol.

The authors provided rame-count ablation studies to address eqYz and hK4s's concerns on frame sampling.

The outstanding concerns are:

Reviewer s340's concern on stronger vision stacks remains. The authors respectfully disagreed, stating that current Video-LLMs are end-to-end trained and cannot have their encoders standalone without full retraining.

Reviewer hK4s's concern on mimimum frame to solve each task remains. The authors explain it as a future extension.

Reviewers concerns on whether synthetic game settings truly enhance real-world understanding is partially addressed by transfer learning data. The concern about the gap between game-based "Active Spatial Cognitive Streams" and "real-world problems" remains.

Reviewer s340's concerns on gpt-4 templates remains.

**Reviewer Scores:**

3 reviewers with negative initial opinions are not likely to raise their ratings towards positive. Reviewer eqYz is intially positive explicitly stated the willingness to keep the original score.

---

### Decision · Program_Chairs · 2026-01-26

Reject